# Symbiont-host interactome mapping reveals effector-targeted modulation of hormone networks and activation of growth promotion

Rory Osborne [1,2,9], Laura Rehneke [3,9], Silke Lehmann [1,4,9], Jemma Roberts[1], Melina Altmann[5], Stefan Altmann[5], Yingqi Zhang[6], Eva Köpff[7], Ana Dominguez-Ferreras [1], Emeka Okechukwu[1], Chrysi Sergaki[1], Charlotte Rich-Griffin[1], Vardis Ntoukakis [1], Ruth Eichmann [3], Weixing Shan [6], Pascal Falter-Braun [5,8,10] ✉ & Patrick Schäfer [3,10] ✉

Plants have benefited from interactions with symbionts for coping with challenging environments since the colonisation of land. The mechanisms of symbiont-mediated beneficial effects and similarities and differences to pathogen strategies are mostly unknown. Here, we use 106 (effector-) proteins, secreted by the symbiont *Serendipita indica* (*Si*) to modulate host physiology, to map interactions with *Arabidopsis thaliana* host proteins. Using integrative network analysis, we show significant convergence on target-proteins shared with pathogens and exclusive targeting of Arabidopsis proteins in the phytohormone signalling network. Functional *in planta* screening and phenotyping of *Si* effectors and interacting proteins reveals previously unknown hormone functions of Arabidopsis proteins and direct beneficial activities mediated by effectors in Arabidopsis. Thus, symbionts and pathogens target a shared molecular microbe-host interface. At the same time *Si* effectors specifically target the plant hormone network and constitute a powerful resource for elucidating the signalling network function and boosting plant productivity.

Plants continuously interact with a plethora of prokaryotic and eukaryotic microbes. While understanding the molecular details of pathogen-host interactions has commanded most attention by researchers, it is increasingly recognised that many microbes benefit their host by improving nutrient acquisition, accelerating growth and boosting pathogen resistance and abiotic stress tolerance[1–5]. In fact, by enhancing plant fitness, symbionts enabled colonisation of the hostile land environment by plants more than 400 million years ago[6]. Ever

[1]School of Life Sciences, University of Warwick, Coventry CV4 7AL, UK. [2]School of Biosciences, University of Birmingham, Edgbaston B15 2TT, UK. [3]Institute of Phytopathology, Research Centre for BioSystems, Land Use and Nutrition, Justus Liebig University, 35392 Giessen, Germany. [4]Laboratory of Biotechnology and Marine Chemistry LBCM, EA3884, IUEM, Southern Brittany University, 56000 Vannes, France. [5]Institute of Network Biology, Molecular Targets and Therapeutics Center, Helmholtz Munich, 85764 Munich-Neuherberg, Germany. [6]State Key Laboratory of Crop Stress Biology in Arid Areas and College of Agronomy, Northwest A&F University, Yangling 712100, China. [7]Institute of Molecular Botany, Ulm University, 89069 Ulm, Germany. [8]Microbe-Host Interactions, Faculty of Biology, Ludwig-Maximilians-University München, 82152 Planegg-Martinsried, Germany. [9]These authors contributed equally: Rory Osborne, Laura Rehneke, Silke Lehmann. [10]These authors jointly supervised this work: Pascal Falter-Braun, Patrick Schäfer. ✉e-mail: pascal.falter-braun@helmholtz-muenchen.de; patrick.schaefer@agrar.uni-giessen.de

since, plants have co-evolved with and relied on beneficial microbes to withstand challenging environments[7–9]. However, while individual mechanisms underlying health-promoting effects have been worked out in some cases, a systems-mechanistic-level understanding of how symbionts interact with plant hosts is lacking. Moreover, beneficial microbes and evolutionary close pathogenic relatives share many host-interacting mechanisms and it remains a fundamental open question how these interactions turn into a pathogenic or beneficial outcome[10].

Irrespective of their different colonisation strategies, the ability of plant pathogenic bacteria, fungi, and oomycetes to establish diseases is based on the secretion of an arsenal of effector proteins for the targeted manipulation of host pathways. Global interactome studies revealed that effectors from the fungal leaf pathogen *Golovinomyces orontii* (*Gor*), the oomycete leaf pathogen *Hyaloperonospora arabidopsidis* (*Hpa*), the bacterial leaf pathogen *Pseudomonas syringae* (*Psy*), and bacterial root pathogens *Ralstonia pseudosolanacearum* (*Rps*) and *Xanthomonas campestris* pv. *campestris* (*Xcc*) interact with *Arabidopsis thaliana* (hereafter Arabidopsis) host proteins with high specificity, while also sharing common target proteins involved in the regulation of plant immunity. This significant interspecies convergence on targeted host protein networks by pathogens of different kingdoms and trophic colonisation strategies suggested the existence of a common molecular pathogen-host interface that is targeted by diverse microbes[11]. In the absence of systematic interactome data for any beneficial microbes, however, it is unclear whether this common interface is indeed specific to pathogens, or whether it might be a universal microbe host interface. We aimed to address this fundamental question.

In addition, the beneficial (mutualistic) fungal root endophyte *Serendipita indica* (*Si*, formerly *Piriformospora indica*) induces a broad spectrum of beneficial effects (e.g. growth promotion) in its host plants[4,12–20]. Thus, *Si* constitutes an important genetic resource for improving crop productivity under changing environments, which remains untapped due to our lack of understanding of many of the underlying mechanisms. Here, we identify 106 candidate effectors from *Si* and mapped the protein contact points of these in their Arabidopsis protein host. Combining this symbiont-host interactome with that of pathogens revealed, in addition to a common host-microbe protein-interaction interface, an over-representation of *Si-specific* targets within the host hormone network. By implementing an *in planta* phenotyping platform, we show that over 80% of *Si* effectors modulate hormone signalling, and that overexpression of hormone-modulating effectors promotes growth in Arabidopsis. Finally, by integrating our interactome and phenotyping data with an updated hormone protein network, we successfully confirm effector-informed hormone functions for hitherto uncategorised Arabidopsis proteins. Our study thus indicates the translational potential of effectors from the beneficial fungal endophyte *S. indica* in assigning proteins to the highly interconnected plant hormone network and in advancing our understanding of the molecular nature of beneficial plant effects.

## Results

### Interactome mapping of *S. indica* effector-host targets
As effectors from beneficial microbes likely mediate plant symbioses and beneficial host effects[21–24], we aimed to identify *Si* effector candidates (SIECs). We performed RNA-seq of Arabidopsis roots colonised by *Si* at early, biotrophic[25] (3 days after inoculation, dai) and late, cell death inducing[25] (10 dai) colonisation stages (Supplementary Fig. 1a, Supplementary Data 1, 8). Employing approved effector identification pipelines[26–28] we found 106 SIECs that met the stringent selection criteria, such as high expression during colonisation, presence of a signal peptide, and absence of transmembrane domains (Supplementary Data 1). Using the heterologous yeast signal sequence trap (YSST) system[29], we experimentally confirmed the functional integrity of the secretion signals for 11 randomly selected SIECs. The YSST yeast strain

is not able to grow in sucrose medium due to a deletion of the *SUC2* gene encoding a secreted invertase. N-terminal fusion of a protein with a functional signal peptide restored SUC2 secretion and yeast growth in sucrose media, as did *SUC2*-deficient cells complemented with SUC2 (lacking endogenous signal peptide) fused to any of the 11 *SIECs* tested (Fig. 1a, Supplementary Fig. 1b). Expression of *SIECs* did not affect yeast growth under control conditions on glucose media (Supplementary Fig. 1c). Confirming that our search criteria revealed secreted effector candidates, the 106 SIECs were forwarded to systematic large-scale SIEC-Arabidopsis protein interaction (interactome) mapping.

We generated an SIEC-host protein interactome network map using a high-quality pipeline that we previously employed to generate plant pathogen networks[11,30], the Arabidopsis Interactome map AI-1[31], and a systematic map of the phytohormone signalling network[32]. SIECs were screened as DNA-binding protein fusions (DB-SIECs) against a library of 12,000 Arabidopsis proteins (12k_space) and 500 additional proteins associated with hormone signalling[32]. After removal of autoactivating DB-SIECs, we uncovered and verified 207 protein-protein interactions (PPI) between 156 Arabidopsis proteins and 33 SIECs (Fig. 1b and Supplementary Data 2). 14 SIECs interacted with only one host protein, while 19 interacted with two or more host proteins (Fig. 1b, c). 115 host proteins were targeted by only one SIEC, while 41 were targeted by more than one (Fig. 1d). This partition pattern of SIEC targeting is consistent with previously observed pathogen effector-host protein interactions[11,30]. To confirm the quality of our dataset experimentally, we analysed six randomly selected interactions between SIECs and host proteins in independent co-immunoprecipitation assays *in planta* (Fig. 1e). After individual optimisation, all interactions could be confirmed, thus further suggesting a high biophysical quality of the data. Subsequent gene ontology (GO) enrichment analysis indicated that SIEC-targeted Arabidopsis proteins function in processes that are known to regulate symbiotic *Si*-Arabidopsis interactions, including defence response (GO:0031347), regulation of metabolism (GO:0009893), regulation of developmental process (GO:0050793) and cellular response to hormone (GO:0032870) (Supplementary Data 1, 3)[25]. Further supporting the validity of the SIEC interactome map (Fig. 1b), 19 GO terms were common between SIEC targets and differentially expressed Arabidopsis genes (DEGs) in our RNA-seq analyses (Supplementary Data 1, 3), including programmed cell death (GO:0012501), response to radical oxygen (GO:1901700), plant organ development (GO:0099402), indolalkylamine biosynthesis (GO:0046219) and response to ethylene (GO:0009723). In both the DEGs and the interactome, enrichments for over 150 biological processes were over-represented ($p < 0.05$) illustrating in both datasets the broad extent to which *Si* modulates host systems. Terms associated with more specific secondary metabolite biosynthesis such as glucosinolate were observed only for the DEGs, although this may reflect the increased depth of RNA sequencing vs. the less exhaustive Y2H screen. Overall, 20% (34/174) and 32% (59/183) of terms were linked to host defences or hormone signalling in the *Si* interactome and DEGs, respectively. This immune targeting by symbiotic *Si* is consistent with previous reports[25,33,34] and supports in addition to the robustness, the biological validity of the dataset.

### Comparative symbiont-host pathogen-host interactome analyses
Next, we aimed to explore the relationship of symbiotic *Si* host targets to those of pathogen effectors. We previously described the convergence of pathogen effector proteins on few functionally important host proteins, which we coined *intraspecies convergence*, i.e. the targeting of host proteins by multiple effectors from one pathogen[11]. We

also described *interspecies convergence*, for the targeting of common host proteins by different and evolutionary distant pathogens (*Pst*, *Hpa*, *Gor*[11,30]), which was later extended to *Ralstonia pseudosolanacearum* (*Rps*) and *Xanthomonas campestris* (*Xcc*)[35]. Notably, we showed

that topological convergence corresponds to biological importance: in infection assays the level of convergence correlated with the frequency of enhanced resistance and enhanced susceptibility phenotypes[11]. Beyond being important for the successful colonisation of diverse

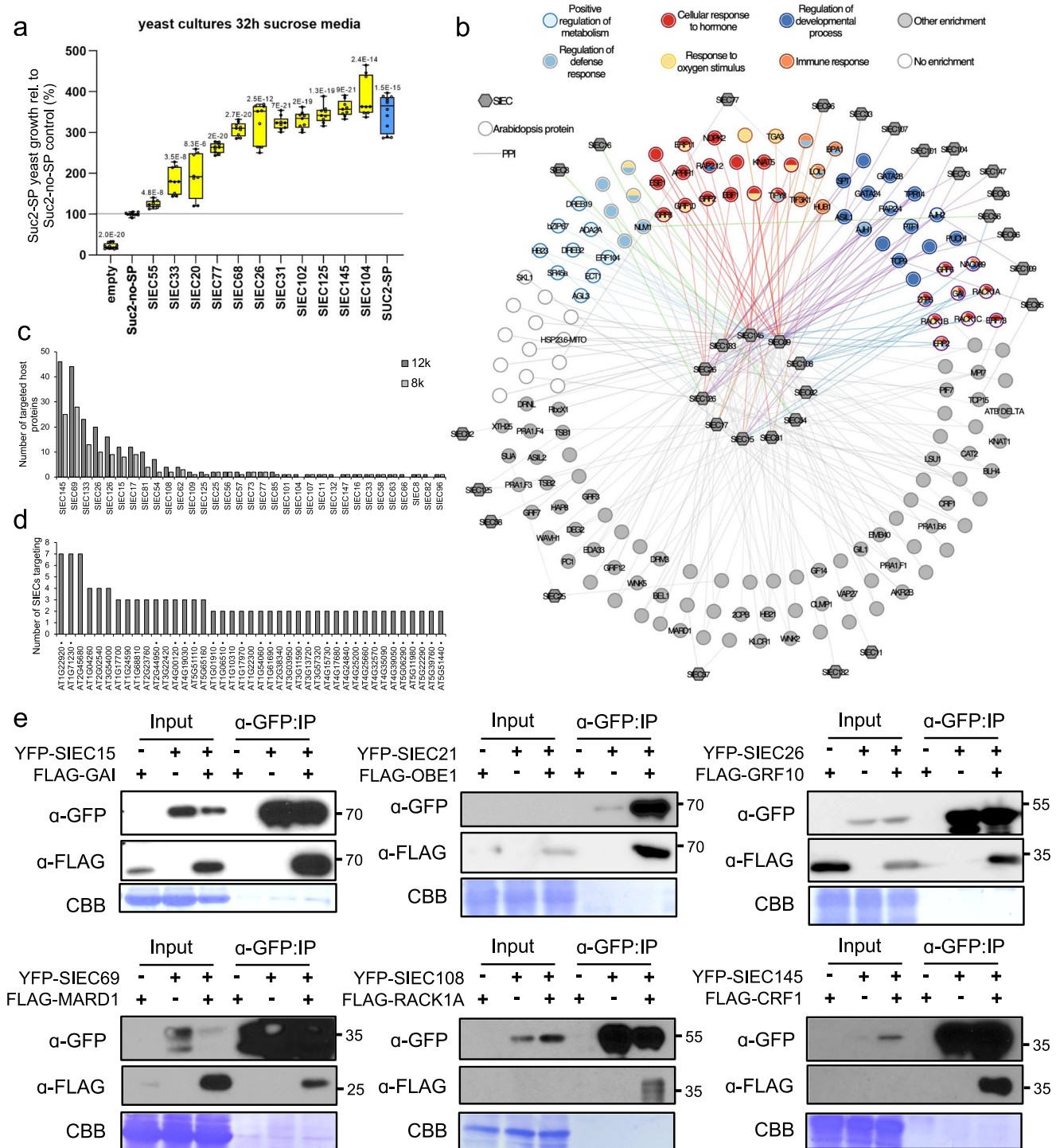

**Fig. 1 | *Serendipita indica* - Arabidopsis interactome map. a** The expression of signal peptide (SP)-containing SIECs fused to yeast SUC2 allows for growth of yeast cells on secretion selection media when compared to expression of SUC2 without SP (no-SP). Empty = untransformed yeast Y02321, SUC2-SP = SUC2 with endogenous SP (blue; positive control). Error bars represent min to max from *n* = 3 biological replicates. Yellow colour indicates significantly growing yeast cultures according to two-tailed, unpaired t-test: numbers above plots indicate *p*-values for significantly different comparisons. See also Supplementary Fig. 1. All box plots indicate minimum to maximum values, the 25th to 75th percentile with lines

indicating the median of the data. **b** Interaction network displaying Arabidopsis proteins (circles) targeted by *Si* effector candidates (SIECs, hexagons), and the gene ontology (GO) term enrichment for these proteins (see Supplementary Data 3 for details). **c** Degree distribution of SIECs interacting with Arabidopsis proteins in the 12k_ (dark grey) and 8k_spaces (light grey). **d** Degree distribution of Arabidopsis proteins that are targeted by at least two SIECs. Dots indicate 8k space plant proteins. **e** Co-immunoprecipitation assays confirm interactions of SIECs with their predicted Arabidopsis target proteins *in planta*. Numbers indicate size (kDa).

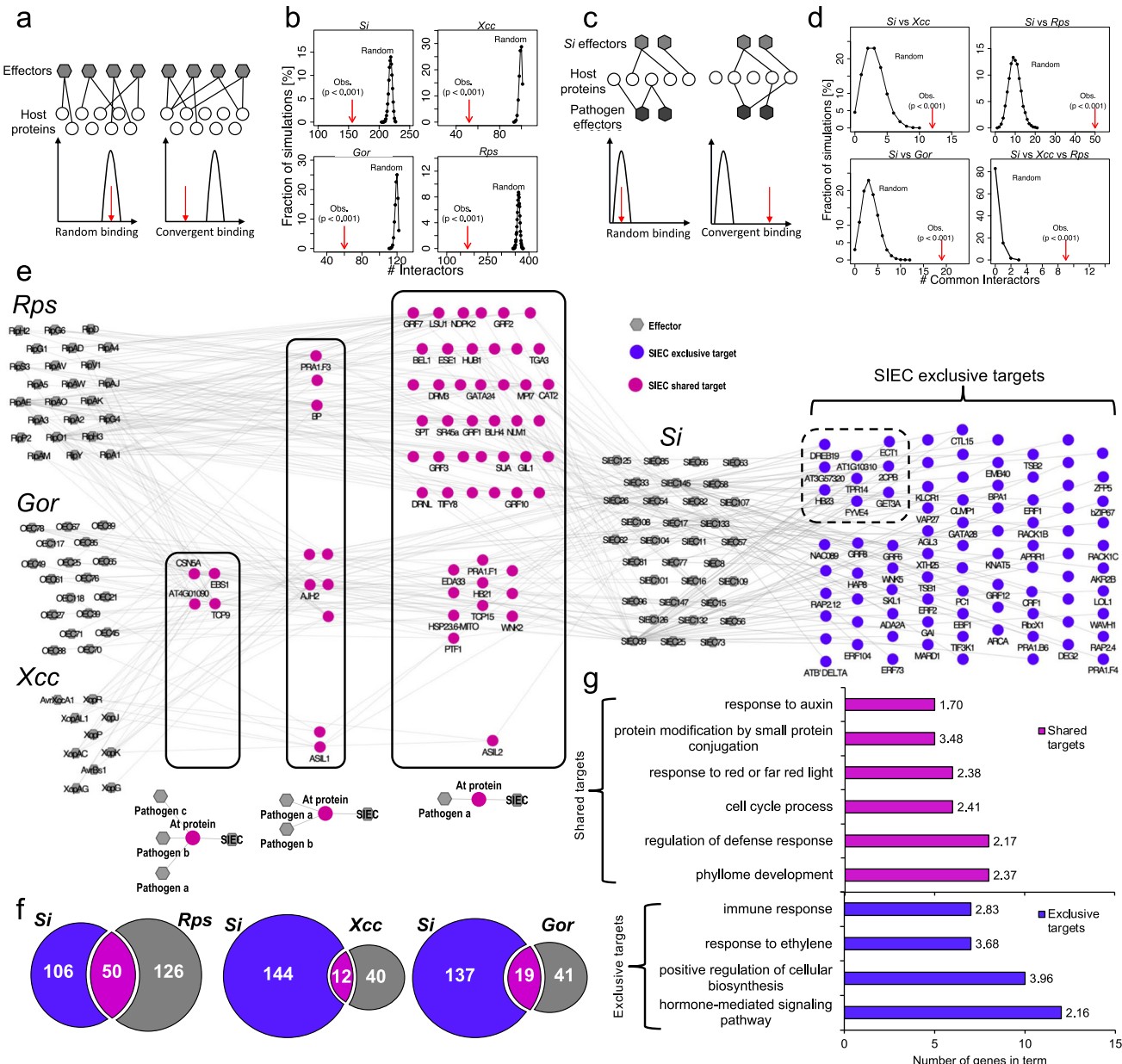

**Fig. 2 | Comparative interactomics (12k_space). a** Network rewiring between effectors (hexagons) and host proteins (circles) to determine the likelihood of random and convergent binding between effectors and host proteins.
**b** Distribution of the number of simulated interactors between effectors from *Si, Xcc, Gor and Rps*, and *Arabidopsis* proteins vs. the observed number (red arrows).
**c** Network rewiring of interactions from effectors of multiple organisms targeting host proteins to determine the likelihood of interspecies convergence.
**d** Distribution of the number of random simulated common interactors between SIECs and effectors of *Xcc, Rps, Gor* vs. the observed number of common interactors (red arrows) in two- and three-way convergence analyses. **e** Classification of

SIEC target proteins as either exclusive to *Si* (blue) or shared with at least one pathogen effector (magenta) from *Rps, Xcc* or *Gor*. Shared nodes are hierarchically displayed according to the number of microbes with interacting effectors. Pathogen exclusive interactions not shown. Dashed inset: convergent targets exclusive to *Si*. **f** Overlap between SIEC and pathogen effector interactions from *Rps, Gor* and *Xcc*. **g** Comparative GO term enrichment analysis showing number of genes represented by terms which were present in exclusive (blue) or shared (magenta, *Rps, Xcc, Gor*) SIEC targets. Numbers at the end of bars represent the -log₁₀ of the adjusted p-value for each term.

pathogens, the convergence targets are linked to population genomic signatures[11] and even conserved as pathogen targets between plant host species[36]. Because of the intimate co-evolutionary link between plants and microbes, we wondered if convergence targets are perhaps not solely important for pathogens, but instead might be universal molecular microbe-host contact points. We therefore examined if SIECs also exhibited intraspecies convergence, and if some of the convergence targets are shared with pathogens[11,35]. Our analysis clearly revealed intraspecies convergence for SIECs (Fig. 2a, b), where the simulated number of unique interactors is higher than the number

observed, indicating that redundant targeting is not exclusive to pathogens but may be a common feature of plant-microbe interactions. For analysing interspecies convergence, we integrated our data with pathogen-host interactions for *Rps, Xcc, Gor, Hpa, Psy* effectors[11,30,35] taking the search space of each experiment into account. The effectors of *Hpa* and *Psy* were tested in an 8k_space (8,000 host proteins)[30], which is a fully contained subset of the 12k_space[11,35] subsequently used for screening *Gor, Rps* and *Xcc*[11,35]. In both search spaces degree-preserving network rewiring revealed clear evidence for interspecies convergence involving SIECs (Fig. 2c, d and

Supplementary Fig. 2a–c) in all possible comparisons, which cannot be explained by random effects. In line with the GO analyses (Fig. 1b and Supplementary Data 3) this indicates that *Si* and pathogens employ effectors interacting with some of the same host proteins to manipulate common processes in Arabidopsis irrespective of evolutionary origin (prokaryote, eukaryote) and the lifestyle (beneficial, pathogenic) of the microbes. For example, 13 SIECs shared four Arabidopsis protein targets with all three pathogens in the 12k search space: TCP9, EBS1, CSN5A, and AT4G01090 (Fig. 2e), all of which have been implicated in plant defence. This analysis also revealed several convergent host proteins which were not targets of pathogen effectors, such as HOMEOBOX PROTEIN 23 (HB23) and DEHYDRATION RESPONSE ELEMENT-BINDING 19 (DREB19), which may be important to *Si* colonisation specifically. Overall, *Si* shared most targets with *Rps* effectors (12k_space) and *Hpa* effectors (8k_space) (Fig. 2e, f and Supplementary Fig. 2e), respectively, which likely reflects the common target tissue (roots) and colonisation strategy (biotrophy with a subsequent cell death phase) of the microbes.

### *S. indica* effectors target the host hormone network

While the convergence analyses revealed shared targeting of SIECs and pathogen effectors, we next explored the extent of SIEC exclusive targeting and the function of these targets. In the 12k_space, *Si* had 87 exclusive targets; in the universally interrogated 8k_space, 76 Arabidopsis proteins exclusively interacted with SIECs (Supplementary Fig. 2d, e). To identify any discriminate functional trends between shared and exclusive SIEC targets we performed a second enrichment analysis with each set of proteins restricted to a reference set of Y2H positive genes (see Methods). As most enriched terms were associated with broad regulatory processes, we focused on annotations that were restricted to <10% of the total number of genes in the reference set (280), and subsequently removed terms for which <5% of the query set were enriched. Positive regulation of cell biosynthesis (GO:0031328), pertaining to increases in cellular anabolism, was the most significant term in the exclusive set, whilst protein modification by small protein conjugation (GO:0070647) was most highly enriched in the shared targets. Hence these distinctions indicate the diverse function of *Si* effectors, among which the exclusive targets might regulate growth via anabolic processes, whilst others suppress host immunity, e.g. through the ubiquitin proteasome system, a central hub already implicated in many plant-pathogen interactions[37–40]. Terms associated with innate immune response (GO:0045087) and regulation of plant-type hypersensitive response (GO:0010363) were enriched in SIEC exclusive targets, but not in the shared target set. Conversely, regulation of defence response (GO:0031347) was enriched in shared targets only. These differences in the enrichment of immunity-related GO terms may reflect incomplete saturation of the pathogen-host screens[41], but could also point to *Si-specific* mechanisms of immune modulation. Intriguingly, *Si*-exclusive host targets were distinctly enriched for GO terms associated with hormone signalling and response to ethylene, whilst shared host targets were enriched for response to auxin only (Fig. 2g and Supplementary Data 3, 4). While the significance of hormones in the establishment of *Si* symbioses is well known[17,25,42], only recently have we begun to understand the relevance of hormone function in *Si*-mediated beneficial effects[19,43–45]. The strong enrichment of hormone-related GO terms for exclusive *Si* host targets suggests an important role for SIECs in operating beneficial host activities. To gain a more comprehensive insight into the interconnection of *Si* and the host hormone network, we refined the SIEC interactome to uncover SIEC-hormone interaction points (SHIPs). We extended this concept of SHIPs from our previous definition of pathway contact points to identify crosstalk in the phytohormone network[32]. After systematically mapping the Arabidopsis phytohormone interactome network, we reported an abundance and functional significance of physical contacts between proteins associated to different phytohormone

signalling pathways, and validated that such contacts point to signalling cross-talk and functional pleiotropy of the involved proteins[32]. In fact, interactions between differently annotated Arabidopsis proteins reliably pointed to unknown functions for at least one of them, often both[32]. To identify SHIPs in this study, we first integrated the SIEC interactome with the systematic portion of the Arabidopsis Interactome 1 (AI-1$_{MAIN}$), and annotated hormone functions in this network using the Arabidopsis hormone database 2 (AHD2)[46,47] and hormone-related GO terms (Fig. 3a). This revealed interactions of SIECs with Arabidopsis hormone proteins that we defined as direct SHIPs (1$^{st}$ degree) and 2$^{nd}$ degree SHIPs, i.e. SIEC-hormone protein interactions via hormone-un-annotated mediators[31] (Fig. 3a–c). Of the 33 SIECs in our SIEC interactome (Figs. 1b and 2e), 20 formed 1$^{st}$ degree SHIPs with 50 hormone-annotated targets, and 7 SIECs formed 2$^{nd}$ degree SHIPs with 62 hormone proteins. 6 SIECs had no 1$^{st}$ or 2$^{nd}$ degree hormone-interaction and their targets remained un-annotated (Fig. 3a, b). The observed frequency of SIEC interactions with hormone proteins was higher than simulation-based random expectation, and more significant when compared to pathogen-effector interactions with the hormone network (Fig. 3d, Supplementary Fig. 3). Given that we were unable to find interactors for 73 of the 106 SIECs, and that 66 SIEC targets remained un-annotated (Fig. 3a), we hypothesised that the interconnection between SIECs and the hormone network was considerably deeper than we had observed. Our findings, the known completeness limitations of large-scale interactome maps[31,48] and the potential for beneficial activities of hormone-targeting SIECs, encouraged us to systematically explore the function of all 106 SIECs in hormone signalling in complementary, functional plant screens.

### Host hormone regulatory functions of *S. indica* effectors

To study functions of the 106 SIECs in hormone signalling we employed our previously developed promoter-based hormone reporters (hereafter *pHORMONE*) for Arabidopsis protoplast assays. These *pHORMONE* reporters are highly specific for each of five hormones (ABA, AUX, CK, JA, SA)[49] which cover a wide range of plant signalling processes, including growth, development, biotic and abiotic stress responses. We were unable to find specific reporters for the remaining hormones. For the functional protoplast screen, we expressed a *LUCIFERASE* (*LUC*) coding sequence using the promoters of respective AUX-, CK-, ABA-, SA- or JA-responsive genes (*pHORMONE::LUC*) together with individual SIECs under the control of the cauliflower mosaic virus *35S* promoter (*35S::SIECs*); *UBIQUITIN10* promoter driven expression of *GLUCURONIDASE* (*pUBQ10::GUS*) was used for normalisation (Fig. 4a). We first conducted a landmark screen followed by a validation screen. In the landmark screen, each of the 106 SIECs x 5 hormone marker combinations were tested in stimulated (respective hormone treatment) and unstimulated (mock) conditions and in two replicates, resulting in 1060 protoplast assays (Fig. 4a–c). Following data normalisation, changes to LUC signals were considered significant if the SIEC altered a reporter more than 2-fold (stronger/weaker) versus empty vector controls. For each SIEC/reporter combination we obtained one "mock ratio" capturing suppressing or inducing SIEC effects on the *pHORMONE::LUC* reporter in the absence of hormone treatment, and a "treatment ratio" to capture suppressing or inducing SIEC effects on *pHORMONE::LUC* activity in the presence of hormone treatment. The product of these ratios was used to quantify the overall effect of each SIEC on the tested hormone pathway. This processing step highlighted SIECs which function synergistically (marker induced or repressed) in both basal and hormone-treated conditions, adding stringency and simplicity to the analysis by revealing robust and continuous effects in the tested conditions (Fig. 4b). Of 530 tested SIEC/hormone marker combinations we detected significant changes in 166 (31%) and of these, 43% (72) were reporter inductions, while 57% (94) were suppressions (Fig. 4b).

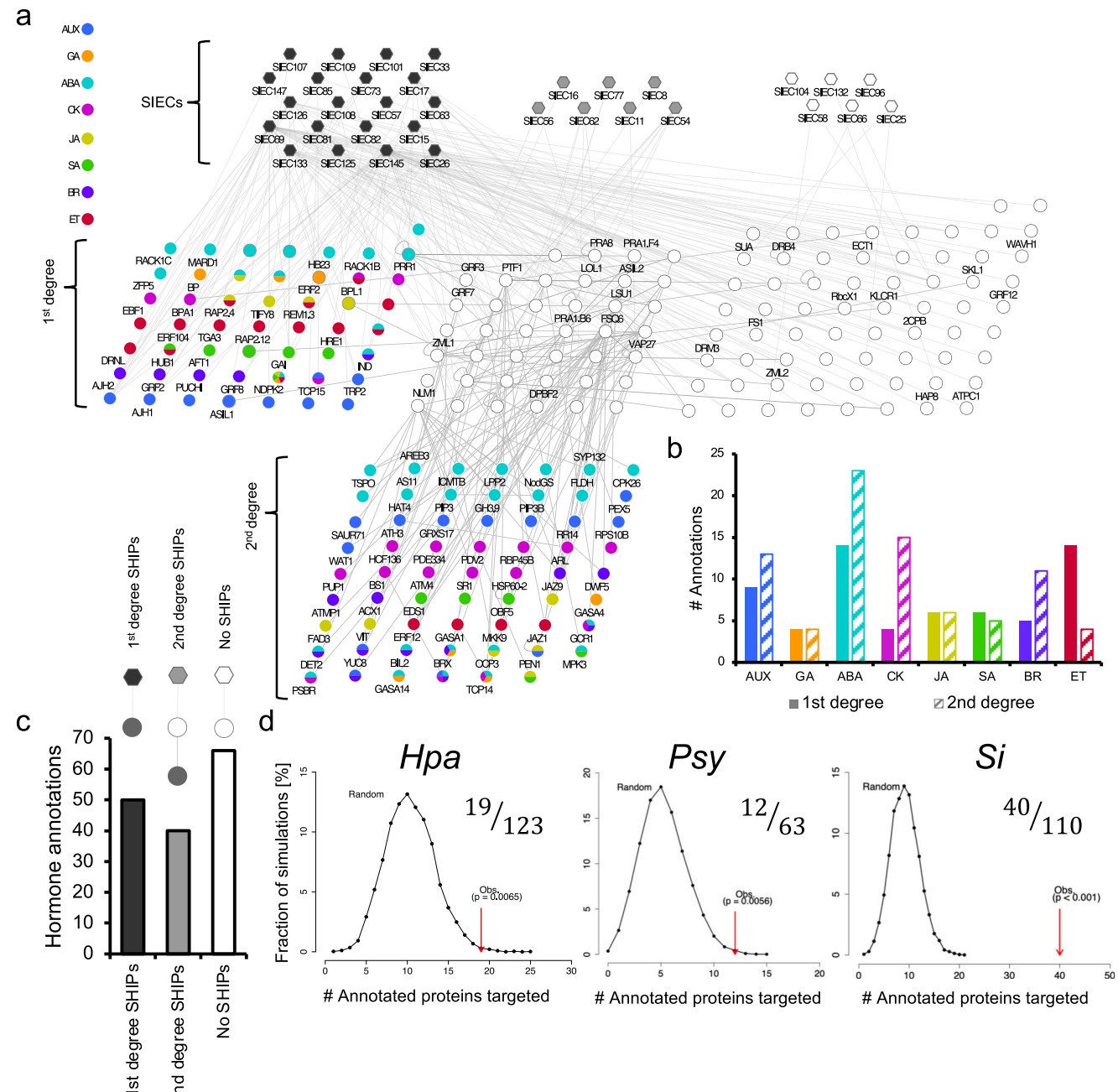

**Fig. 3 | SIECs target hormone pathways in Arabidopsis. a** Combined phyto-hormone annotations from the Arabidopsis Hormone Database v2 and Gene Ontologies mapped onto SIEC target proteins. 1st degree SIEC hormone interacting points (SHIPs) interact directly with respective SIECs (dark grey). 2nd degree SHIPs are not themselves annotated but form secondary interactions with other anno-tated proteins within AI-MAIN1. **b** Summary of hormone annotations in 1st and 2nd degree SHIPs. For bar colours see legend in **a**. **c** Number of SHIPs in the SIEC interaction network. Legend: Grey and white nodes represent annotated and un-annotated proteins respectively. **d** Frequency distribution of the number of annotated proteins targeted by effectors (8k space) in random networks vs the observed number for *Hpa*, *Psy* and *Si*. Inset numbers represent the # of annotations (top) vs the total # of unique effector targets (bottom).

The highest number of SIEC-dependent reporter changes was observed for the AUX and SA reporters (37 SIECs each), followed by ABA (36), JA (35), and CK (21) (Fig. 4b). Except for the SA reporter, suppressions were more prevalent than inductions. Overall, 86 of the 106 SIECs (80%) changed at least one of the 5 hormone pathways in the landmark screen. An independent validation screen (Fig. 4d) con-firmed all SIECs for JA, 90% for AUX, 70% of CK and 60% of ABA effects. Only 10% of SIEC effects on the SA reporter were validated (Fig. 4d) likely due to the overall low, albeit highly specific, LUC signal of the SA reporter. Consistent with the observed target-specificity of SIECs in our SIEC interactome map (Fig. 1b, c), we detected a high SIEC-pathway specificity in our protoplast screen, as 39 of the 86 hormone-modulating SIECs caused significant changes in only one of the five tested hormonal pathways (Fig. 4c). In addition to these very specific SIEC effects, we also identified six broad range hormone signalling modulating SIECs that changed at least four of the five tested markers (Supplementary Data 5). Given the complexity of the host hormone network, and that small perturbations in the plant metabolism (sig-nalling/biosynthesis/perception) can alter hormone regulated gene expression, particularly via crosstalk, it is important to consider whe-ther measured changes in marker expression represent direct activi-ties, or reflect the response by host cells to upstream activities of each

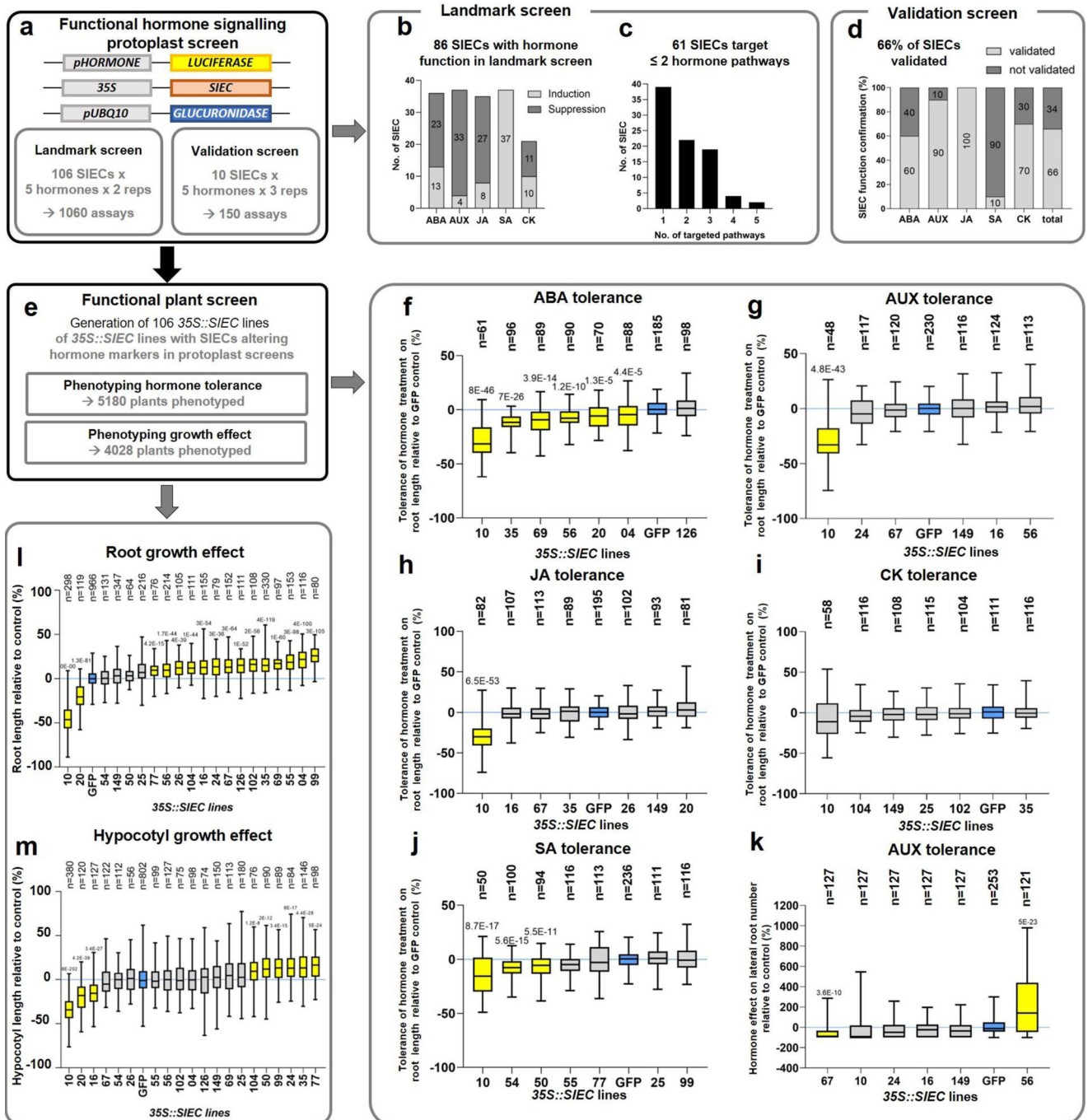

**Fig. 4 | Phenotyping of hormone-related effects of SIECs in protoplasts and whole plants. a** Workflow for annotating hormone functions of SIECs in functional Arabidopsis protoplasts landmark and validation screens. Protoplasts were transformed with an SIEC overexpression construct (*35S::SIEC*), hormone markers consisting of promoters of hormone-specific genes (see Materials & Methods for details) fused to LUCIFERASE (*pHORMONE::LUCIFERASE*) and a *pUB-Q10::GLUCURONIDASE* construct for data normalisation. **b** Number of SIECs that induce or suppress tested hormone pathways in the landmark screen with all 106 SIECs tested against five hormones (1060 assays in total). **c** Specificity of SIECs in modulating hormone pathways. **d** Validation screen for the top ten SIECs (based on 106 SIEC from the landmark screen) on hormone marker regulation (150 assays in total). **e** Outline for functional plant screen to phenotype the effect of SIECs on altering hormone tolerance or growth in whole plants. Hormone tolerance of *35S::SIEC* lines compared to *35S::GFP* control plants (blue) for ABA (**f**), AUX (**g, k**), JA

(**h**), CK (**i**), and SA (**j**) (5180 plants in total). Error bars represent min to max from 3 biological replicates. Yellow colour indicates significant growth differences compared to *35S::GFP* plants. Statistical significant difference was calculated using two-tailed, unpaired t-test: numbers above plots indicate *p*-value for significantly different comparisons. All box plots indicate minimum to maximum values, the 25th to 75th percentile with lines indicating the median of the data. **l, m** *35S::SIEC* lines with altered primary root or hypocotyl length when compared to *35S::GFP* control plants (blue). Error bars represent min to max from at least 2 biological replicates (4028 plants in total). Yellow colour indicates significant differences compared to *35S::GFP* plants. Statistical significant difference was determined by two-tailed, unpaired t-test: numbers above plots indicate *p*-value for significantly different comparisons. All box plots indicate minimum to maximum values, the 25th to 75th percentile with lines indicating the median of the data.

effector. In either case this analysis gives strong evidence toward which pathways are modulated in the presence of each SIEC, particularly as all *pHORMONE::LUC* constructs tested were highly specific toward their respective hormones[49], and many effector candidates only affected one marker.

To validate the protoplast-detected SIEC signalling functions in whole plants, we stably expressed all 106 SIECs in Arabidopsis seedlings (Fig. 4e) and confirmed *SIEC* expression by quantitative real time-PCR (qRT-PCR) (Supplementary Fig. 4). We phenotyped primary root length reduction (relative to *35S::GFP* plants) upon hormone treatment as a proxy for hormone-tolerance of the lines (Fig. 4f–j, Supplementary Data 6 and Supplementary Fig. 5). Analysing 5180 *35S::SIEC* plants (~860 plants/hormone treatment), we observed phenotypes for JA, SA, ABA and AUX, but not CK (Fig. 4f–j). Interestingly, almost all observed phenotypes indicated SIECs reduced tolerance to respective hormones. For example, 6 of the 7 *35S::SIEC* lines showed diminished ABA tolerance as indicated by increased root growth attenuation (Fig. 4f). We further analysed lateral root number (LRN) for AUX and CK treatment as a second phenotype regulated by both hormones, and anthocyanin production under constant light for CK tolerance (Fig. 4k and Supplementary Fig. 7g, h). While SIEC effects were not detected for CK, reduced and enhanced lateral root formation were observed for the AUX-treated lines *35S::SIEC67* and *35S::SIEC56* (Fig. 4k) respectively. *35S::SIEC10* plants showed reduced root growth on control media which might affect its performance after hormone treatment (Fig. 4l). SIEC10 strongly induced expression of all 5 hormone markers in protoplast screens (Supplementary Data 5), suggesting this effector might influence multiple fundamental processes in plants. We were able to confirm an *in planta* function for 4 out of 5 hormones. Whilst as expected, the direction/amplitude by which an effector modulated each hormone marker in protoplasts did not clearly correlate with the direction of hormone responsiveness at the root level (Supplementary Fig. 6), the confirmation of SIEC effects in whole plants indicated the specificity and suitability of the protoplast assay in analysing and detecting hormone functions of SIECs. This is particularly relevant for hormones and their involvement in a multitude of processes and response reactions of plants. In our study we focused on two root phenotypes and additionally anthocyanin content for cytokinin treatment. Thus, we did not examine all possible traits altered by SIEC expression and hormone treatment. Still, we were able to confirm 39% of the SIECs with effects on hormone signalling, functionally validating the dramatic impact of SIECs on the phytohormone signalling network.

## *S. indica* effectors promote growth in Arabidopsis

Given that *Si*-mediated benefits involve hormone signalling[19,43–45], we wondered if individual SIECs supported *Si* mediated phenotypes, such as growth promotion in Arabidopsis. We phenotyped 20 *35S::SIEC* lines that previously showed altered hormone activity in protoplasts, using primary root and hypocotyl length as a proxy for growth promotion in below- and above-ground tissue. These included 3 SIECs which modulate the response to the growth hormones AUX and CK, 3 SIECs affecting responses to the abiotic stress hormone ABA, 6 SIECs altering the responses to JA and SA, and 8 SIECs with overlapping effects on multiple hormones. Overall, 16 of the 20 (80%) tested SIECs impacted growth control of Arabidopsis roots, and 9 altered hypocotyl length (Fig. 4l, m). A majority of 14 SIECs promoted root growth, whilst only two SIECs impaired it; 6 promoted hypocotyl length, whilst 3 suppressed it. Intriguingly, not all lines with enhanced root length produced longer hypocotyls, and vice versa, suggesting in some cases highly specific physiological activity of SIECs. Thus, even individual SIECs can confer well-known benefits of *Si* symbiosis on host plants. Considering the tight connection of SIECs to the hormone-signalling network it can be anticipated that many other phenotypes are mediated by individual effectors.

## *S. indica* effector-based informing of the host hormone network

The complex and integrated nature of plant hormone signalling networks challenges mapping of protein function to specific hormone pathways[32]. We have previously demonstrated that physical interactions among differently annotated signalling proteins can reliably inform on novel functions of one or even both partners in the respective other pathway[32]. Based on the SIEC effects on hormone pathways, we thus wondered if SHIPs could be used similarly to identify unknown hormone functions of their Arabidopsis targets. To this end, we combined our SIEC-phytohormone protein network (Fig. 3a) with the SIEC protoplast phytohormone assay data (Fig. 4a–d) to assign functions to Arabidopsis proteins according to their interactions with hormone-modulating SIECs. 99 Arabidopsis proteins and their 19 SIEC interactors fit these criteria and were used to generate the Functionally Informed Symbiont-Plant Interaction Network (FI-SPIN, Fig. 5a). We applied the concept of SHIPs to define the different SIEC-hormone protein interactions. 5.6% and 24.4% of edges within FI-SPIN were classified as SHIPs, where SIEC-inferred hormone assignments of Arabidopsis targets match published Arabidopsis protein functions (type I) or do not match (type III), respectively (Fig. 5b). A majority of 70% of SIEC-Arabidopsis target combinations involved un-annotated Arabidopsis proteins, which we refer to as type II SHIPs (Fig. 5b). To elucidate the accuracy of these SHIP-based hypotheses regarding the Arabidopsis hormone network, we chose ten SIEC-interacting Arabidopsis proteins covering type I-III SHIPs (Fig. 5c, Supplementary Fig. 7). Our selection included ASIL1 and XTH25 for type I SHIPs; EF-HAND, DRB4, KAKU4, NAC089, AT3G29270 /RING/U-BOX, and TCP9 for type II SHIPs; and CHLADR and ZFP5 for type III SHIPs (Fig. 5c). We subsequently phenotyped T-DNA mutants lacking respective SIEC-interacting proteins (Supplementary Fig. 7a–j) for altered hormone tolerance by assessing primary root length and LRN (Supplementary Data 6) (~530 mutant plants/treatment, 2643 plants in total), as well as by quantifying respective *pHORMONE::LUC* activities in mutant protoplast assays (Fig. 5d–m).

Among the type II mutants, *ef-hand* and *tcp9* mutants exhibited wild type phenotypes in JA and ABA assays (Fig. 5g, i and Supplementary Fig. 8a). However, in accordance to altered JA tolerance of EF-HAND and TCP9-interacting SIECs in protoplasts assays (Fig. 5c), protoplasts of both mutants revealed higher JA marker expression (*pJA-Z10::LUC*) in the absence of JA treatment, suggesting an elevation of basal JA signalling in both mutants (Fig. 5m). Consistent with the altered AUX responsiveness of protoplasts expressing ASIL1-interacting SIEC107, the AUX marker (*pGH3.3::LUC*) was repressed in *asil1* mutant protoplasts and LRN was enhanced in *asil1* mutants in the presence, but reduced in the absence of AUX (Fig. 5f, k and Supplementary Fig. 8c). In line with the AUX induction by DRB4-interacting SIEC96 (Fig. 5c), *drb4* mutants produced longer roots and more lateral roots under AUX treatment (Fig. 5e, f and Supplementary Fig. 8a, c). This suggests a previously unknown AUX function of DRB4. As detected for interacting SIEC69, *zfp5* mutant protoplasts altered ABA marker expression (*pRD29A::LUC*) upon ABA treatment. In addition, *zfp5* mutant seeds exhibited a higher germination rate in the presence, but not absence of ABA, indicating a previously unknown ABA function of ZFP5 (Fig. 5h, i, l and Supplementary Fig. 8b). *nac089* mutant protoplasts were reduced in JA marker expression, and, consistent with its interactors SIEC56 and SIEC62, mutant plants showed a reduced JA and SA responsiveness (Fig. 5d, g).

Thus, of the ten candidate proteins, mutants for seven (70%) showed altered hormone responsiveness in the root and/or protoplast assays as predicted by our FI-SPIN (Fig. 5a, c). This indicates that indeed SIECs are helpful to identify hitherto unknown functions of Arabidopsis proteins in hormone signalling, and to disentangle highly interconnected pathways. More importantly, these experiments demonstrate the reliability and integrated power of FI-SPIN. Our SIEC ORFeome and the highly validated FI-SPIN network will be a powerful

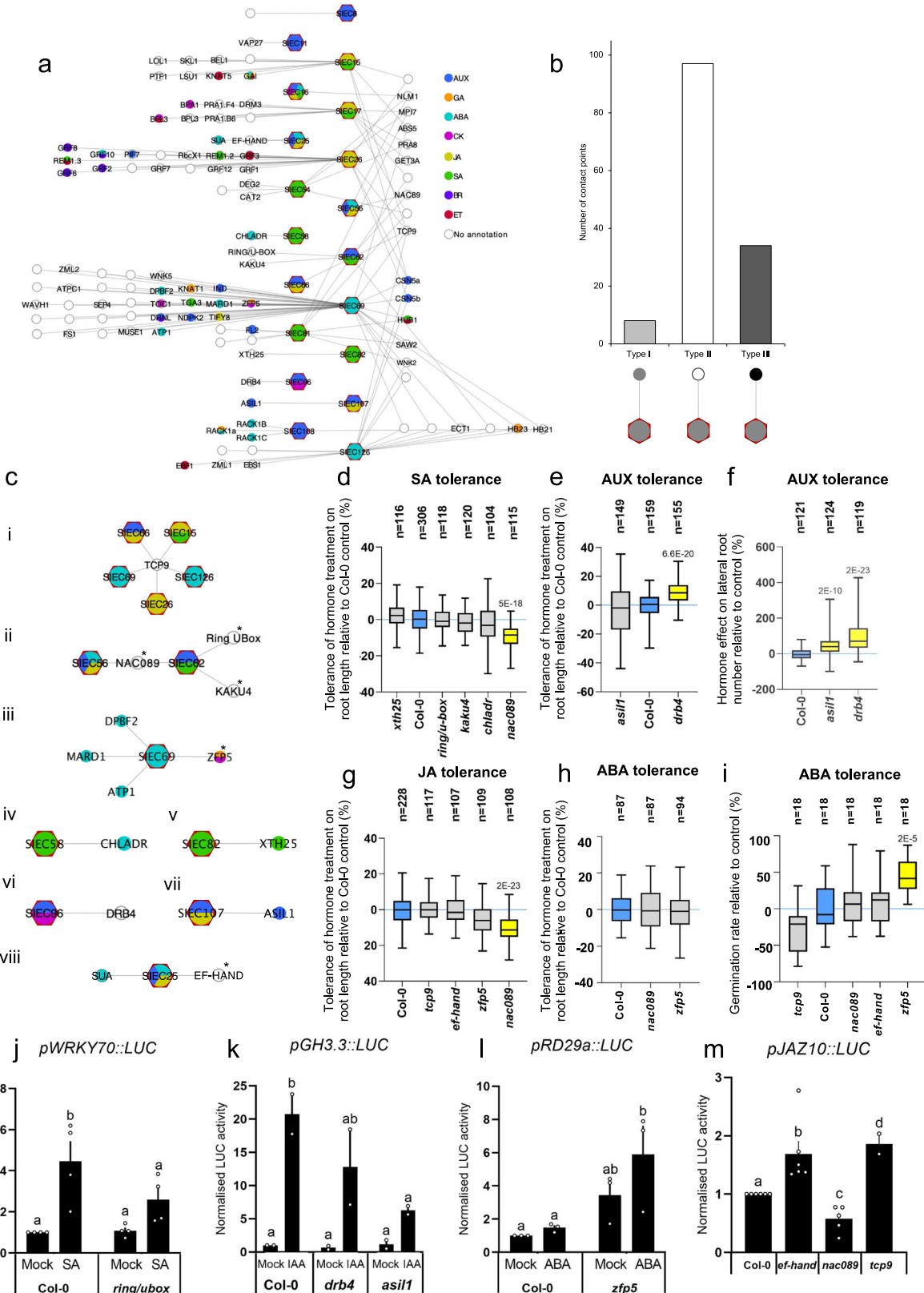

resource to advance the understanding of specific effectors and of the molecular basis of symbiont-host interactions in general.

## Discussion

In co-evolving with plants, microbes have developed highly specific ways of manipulating their hosts for colonisation, nutrient acquisition and even reproduction with a wide range of outcomes for the host. Effectors were first identified in pathogens as secreted proteins that counter plant immune mechanisms. The diversity of effector functions was instrumental both for understanding plant immunity at the molecular level[50,51] and for the development of crops with novel resistance traits[52,53]. However, the understanding of plant interactions with

**Fig. 5 | Function-based informing of the *Si*-Arabidopsis interactome.**
**a** Functionally informed symbiont-protein interaction network (FI-SPIN) showing SIECs which changed at least one hormone pathway in Arabidopsis protoplast (landmark) screen (Fig. 4a–d). Host targets were annotated using AHDv2.0 with additional information from GO terms. Node colours indicate assignment to hormone pathways (top right). **b** Quantification of SIEC-hormone interacting points (SHIPs) in FI-SPIN with type I SHIPs (matching hormone annotation of SIEC and targeted Arabidopsis protein), type II SHIPs (no annotation of SIEC-targeted protein) or type III SHIPs (mismatching hormone annotation of SIEC and targeted Arabidopsis protein). **c** Subnetworks (i-viii) extracted from FI-SPIN (**a**), indicating SIEC-targeted Arabidopsis proteins. T-DNA mutants of these genes were evaluated for phytohormone phenotypes based on their interacting SIEC annotation. Where subnetworks contained multiple Arabidopsis proteins. * Indicates the selected target. Node colours indicate assignment to hormone pathways (top right in **a**). **d–i** Phenotyping of Arabidopsis T-DNA insertion mutants lacking SHIP targets (see **c**) for altered responses to ABA, AUX, JA and SA. Error bars represent min to max from 3 biological replicates. Yellow colour indicates significant differences in hormone sensitivity compared to Col-0 plants (in blue). Statistical significant difference was calculated using two-tailed, unpaired t-test: numbers above plots indicate $p$-value for significantly different comparisons. All box plots indicate minimum to maximum values, the $25^{th}$ to $75^{th}$ percentile with lines indicating the median of the data. **j**–**m** Normalised *pHORMONE::LUC* data in protoplasts of selected Arabidopsis T-DNA mutants relative to Col-0 for *pWRKY70::LUC* (SA), *pGH3.3::LUC* (AUX), *pRD29A::LUC* (ABA) and *pJAZ10::LUC* (JA) with and without hormone treatment. Error bars represent standard errors from at least $n = 2$ biological replicates. Letters indicate significant differences as calculated by ANOVA and Tukey test after normalisation between biological replicates. Data are presented as the mean +/− the SEM.

beneficials and symbionts lags behind that of pathogens, and it is still unclear what exactly determines the outcome of an interaction, e.g. can plants categorically discriminate between pathogenic and beneficial microbes. While some plants possess specific plasmamembrane-localised receptors (such as Nod Factor Receptor 1 (LjNFR1) and LjNFR5 in *Lotus japonicus*) to recognise symbionts and facilitate their accommodation, it is unclear how widespread such specific recognition is among plant species[54] given the diversity of host-beneficial plant-microbe combinations. Moreover, these receptors and the defined set of "common symbiosis genes" required for symbioses in legumes with arbuscular mycorrhizas and N-fixing rhizobia are not required for *S. indica* colonisation[55]. At the same time, all microbial symbionts display microbe-associated molecular patterns that are perceived by pattern-triggered immunity (PTI) receptors. To overcome the effective root immune system, many beneficials, including *S. indica*, employ effector-based strategies analogous to pathogens. In addition to manipulating host immunity at initial interaction stages however, mutualists like *Si* might also employ effectors to elicit effects that are beneficial for their host plants[6,25,56]. The extent to which mutualist and pathogen effectors share host targets and possible functions on one hand, and how symbiont effectors contribute to beneficial effects on the other, are fundamental questions we set out to address.

*Si* possesses a diverse repertoire of effectors[33,57–59], which we used for systematic interactome network mapping. As the same platform was previously used to map plant targets of pathogen effectors[11,30], this allowed us to analyse shared and specific targets of pathogen and symbiont effectors without distortion by differences due to the search space and screening pipeline. Notably, we found that *Si* effectors converge on some of the same host targets that we previously identified as targets for evolutionary distant pathogens, which have important functions in the infection process. The observation that *Si* effectors converge on some of these shared targets suggests that these host proteins are important, independent of the specific outcome of the microbe-host interaction, and constitute key mediators of plant-microbe interactions. Furthermore, our analysis revealed several convergent *Si* effector targets which did not interact with pathogen effectors. Given that we previously reported a correlation between the likelihood of a colonisation phenotype and the number of intra- and cross-species effectors Arabidopsis proteins interacted with[30], these targets will be highly valuable in elucidating the balance between enhanced host fitness and colonisation of *Si* in the future. In addition to *Si* exclusive targets, we found substantial overlap in targets with different pathogens. This observation was most pronounced with the bacterial root pathogen *Rps* and hemi-biotrophic oomycete *Hpa*. Similar targeting thus appears to be driven by similarities in lifestyle and tropism rather than evolutionary proximity; *Si* and *Rps* share a broad host range and target tissue[60,61]. The relatedness of colonisation strategy (biotrophic followed by cell-death associated growth phase for *Si*), in turn, might determine effector target overlaps with hemi-biotrophic *Hpa*.

Beneficial effects mediated by fungi or rhizobacteria often involve changes in plant hormone signalling[62] and previous studies have linked the *Si*-Arabidopsis symbiosis and beneficial effects in growth and defence to modulated hormone signalling[18,24,58,61,63]. It was therefore noteworthy that *Si* effector-targeted proteins are enriched for unique hormone signalling proteins, considerably more when compared to pathogen effector targets, suggesting a broad and deep modulation of the pleiotropic plant hormone signalling network by the symbiont. Our functional screens confirmed this tight connection as 86 out of 106 tested SIECs changed growth and defence hormone marker activities in protoplast assays, many of which were confirmed *in planta*. Whether these changes to hormone marker expression are linked directly to effector function, or are the result of pleiotropy in the plant hormone network, this clearly supports previous observations that irrespective of lifestyle, microbes utilise effectors to target hormone pathways, including those associated with growth and development (BR, CK, AUX), as well as defence (SA, JA, ET)[64–66]. Our analysis suggests this might be particularly important for the balancing of symbiosis in host-microbe interactions, where in addition to the suppression of host defences the activation of beneficial host effects requires manipulation of the hormone network. Given the diversity of SIEC-targeted pathways and the multifunctionality of phytohormone signalling, this intense targeting may be exploitable for biotechnological applications.

However, we also noticed that in many cases the experimentally observed hormone signalling changes could not be directly linked to the annotated hormone functions of SIEC-targeted proteins. We recently demonstrated that such mismatches often result from the pleiotropy of the hormone-signalling network and incompletely characterised protein functions, and that physical interactions reliably identify new protein functions in this network[32]. Thus, to identify indirect links and facilitate hypothesis development about previously unrecognised hormone-related functions of Arabidopsis proteins, we integrated the systematic AI-1$_{MAIN}$ host-interactome with our SIEC-interactome and the protoplast data. The analyses demonstrate the involvement of ER stress-associated NAC089[67] and its interactors SIEC56 and SIEC62 in JA and SA signalling, and an AUX-signalling function of SIEC96 and its target DRB4, a protein so far only known for its role in antiviral defence responses via RNA silencing[68]. Overall, a previously undescribed function in specific hormone signalling pathways could be shown for 7 of the 10 tested SIEC-target mutants, which not only supports our interpretation, but further demonstrates the immense potential of our resource to serve as a reliable basis for advanced hypotheses.

Our study thus provides a deeper insight into *Si* connection to the hormone network, helps assign new hormone-related activities to Arabidopsis proteins, and indicates the usefulness of effectors in decoding highly interactive protein networks such as the hormone system. It further advanced our understanding of the molecular mechanism of *Si*-mediated plant fitness. Many symbiotic plant microbes take part in nutrient exchange at the fungal-plant interface

by providing the host with additional nutrients such as phosphate in exchange for carbon[69–71]. Although some evidence suggests that *Si* behaves in a similar fashion[72–74], systemic benefits associated with the fungus could not be explained simply by an improved nutrient availability. Our findings that SIECs can promote root length demonstrate the potency of SIECs as keys to uncover fundamental processes in plants. Just as pathogen effectors have helped in deepening our understanding of defence signalling in plants, effectors from beneficial organisms, such as *Si*, can help us in identifying beneficial plant traits, and in applying them in the next generation of crops toward an improved fitness under changing environments.

## Methods

### Experimental model and subject details
All Arabidopsis mutants and transgenics are in the Col-0 accession (wild type, WT). *A. thaliana* genetic materials including *xth25*, *asil1-1*, *nac089* and *kaku4-2* were described previously[75–79], except all *35S::SIEC* lines, which were generated in this study, and the *chladr*, *drb4*, *tcp9*, *AT3G29270* (*Ring/U-Box protein*), *zfp5* and *ef-hand* mutants, which were described in this study. Growth conditions for specific experiments are given below in the Arabidopsis growth and transgenic lines section.

### Method details
**Arabidopsis growth and generation of *35S::SIEC* lines.** Plants were grown in vertical squared petri dishes on sterile ½ strength Murashige & Skoog (MS) medium with 0.7% agar or on *Arabidopsis thaliana* salts (ATS) media[80] [5 mM KNO$_3$, 2.5 mM KH$_2$PO$_4$ buffered with 2.5 mM K$_2$HPO$_4$ to pH 5.5, 2 mM MgSO$_4$, 2 mM Ca(NO$_3$)$_2$, 70 μM H$_3$BO$_4$, 50 μM FeEDTA, 14 μM MnCl$_2$, 10 μM NaCl, 1 μM ZnSO$_4$, 0.5 μM CuSO$_4$, 0.2 μM NaMoO$_4$, 0.01 μM CoCl$_2$ and 0.45% (w:v) Gelrite® (Duchefa Biochemie, Netherlands)], in short day conditions with an 10–12 h light (60–120 μmol m$^{-2}$ s$^{-1}$), at 22 °C in a growth cabinet 8 h Before genotyping, plants were grown in ökohum® Anzuchterde mixed with Celaflor Careo® granules (1.5 g in 1 l soil) in a 10 h light (100 μmol m$^{-2}$ s$^{-1}$), 14 h dark cycle at 22 °C. T-DNA insertion mutant lines for putative *Si* effector target genes were acquired from the Nottingham Arabidopsis Stock Centre (NASC): *chladr* (AT3G04000, SALK_068469C), *xth25* (AT5G57550, SALK_204573C), *drb4* (AT3G62800, SAIL813H11), *asil1-1* (AT1G54060, SALK_124095C), *tcp9* (AT2G45680, SALK_201398C), *nac089* (AT5G22290, SALK_201394C), *ring/u-box* (AT3G29270, SALK_082480), *kaku4* (AT4G31430, SALK_076754C), *zfp5* (AT1G10480, SALK_113106C) and *ef-hand* (AT5G28900, SAIL_891_B10). Homozygous integrations of T-DNAs and reduced target gene expression were confirmed by genotyping PCR and qRT-PCR, respectively (see below).

*35S::SIEC* lines were generated by *Agrobacterium tumefaciens*-mediated transformation of expression constructs into Arabidopsis Col-0 wild type plants. SIEC ORFs (see below) were cloned into the Gateway compatible pEarleyGate201 vector[81] and transformed into *A. tumefaciens* strain GV3101. After Arabidopsis transformation, T1 (in soil) and T2 (on agar plates) plants were selected by 150 and 10 μg ml$^{-1}$ Basta treatment, respectively, and genotyped using PCR. Overexpression of SIECs was confirmed by quantitative real-time PCR (qRT-PCR, see below). Three independent lines were generated for each of the 106 SIEC.

### *Serendipita indica* (*Si*) cultivation and treatment
*Si* wild-type strain DSM11827 (Leibniz Institute, Braunschweig, Germany) was grown on CM agar [per litre; 20 g glucose, 2.4 g NaNO$_3$, 2 g peptone, 1 g casamino acids, 1 g yeast extract, 600 mg KH$_2$PO$_4$, 200 mg MgSO$_4$•7H$_2$O, 200 mg KCl, 6 mg MnCl$_2$•4H$_2$O, 2.65 mg ZnSO4•H2O, 1.5 mg H$_3$BO$_3$, 0.75 mg KI, 0.13 mg CuSO$_4$•5H$_2$O, 2.4 ng Na$_2$MoO$_4$•2H$_2$O, 1.5% (w:v) agar] for at least 6–8 weeks in the dark at 25 °C. *Si* spore suspension was prepared by adding H$_2$O 0.02% (v:v) Tween-20 to mature *Si* plates, and scraping off spores and mycelium using a sterile cell scraper. The material was sonicated for 5 min at 4 °C

and filtered through two layers of miracloth. Spores were then centrifuged at 2,200 g and 4 °C for 7 mins, and washed in H$_2$O 0.02% (v:v) Tween-20 three times. Spores were counted using a Fuchs-Rosenthal haemacytometer and made up to 500,000 spores ml$^{-1}$. Arabidopsis seedlings were grown in squared petri dishes on ATS media for 9 days, and then treated with 1 ml per plate *Si* spore suspension. Control plants were treated with 1 ml per plate H$_2$O 0.02% (v:v) Tween-20 (mock). Roots were harvested 3 and 10 days after inoculation and flash frozen in liquid N$_2$.

### In silico identification of SIECs and cloning
Flash frozen mock and *Si* treated Arabidopsis roots were ground in liquid N$_2$ and total RNA was extracted from two biological replicates using TRIzol (Invitrogen)/chloroform. RNA was purified using the RNeasy Plant Mini Kit (Qiagen), including on column Dnase digestion. Library preparation was performed using the TruSeq RNA sample prep v2 kit (Illumina) and raw reads were generated using Illumina HiSeq (50 million reads per sample).

Raw reads were filtered using FastQC[82] and aligned to the Arabidopsis genome (TAIR10) using Bowtie2[83]. SIECs were identified by aligning reads, which did not align to TAIR10, to the *Si* genome[59]. The 852 SIECs identified were then filtered based on completeness of sequence, upregulation during colonisation[84], presence of a signal peptide (SignalP[26]), lack of transmembrane domains (TMHMM[27]) and presence of functional domains (Pfam, ScanPROSITE). The 106 SIECs that remained after filtering were synthesised without signal peptide in the Gateway compatible vector pENTR221 (Life Technologies). For Arabidopsis differential gene expression analysis, mapped read counting was performed using Htseq-count[85]. Differentially expressed genes (DEGs) were identified using DESeq2 after normalisation using default parameters with a log2 fold change cut off of +/−0.6 and an adjusted *p*-value of <0.05 after Benjamini-Hochberg correction.

### GO term enrichment analysis
Gene ontology enrichment analysis was conducted in R studio using the TopGO package[86]. For analysis of proteins identified in protein interaction networks, the reference library of genes is described as Y2H-positive, or genes which showed interaction in AI1-Main[31], PPIN-1[30], PPIN-2[11] and the *Si*, *Rps* and *Xcc* interaction networks. For comparative GO enrichment, GO filtering was implemented to increase specificity and reduce redundancy. GOs were removed if >10% of the total number of genes in the Y2H-positive set (280) were annotated, and if the total number of genes in each term represented <5% of the query set. To calculate GO enrichment of DEGs after *Si* inoculation of Arabidopsis roots, all expressed genes in the Arabidopsis reference genome (TAIR10) were used as a reference library. We used the function runTest to calculate both Fisher and Kolmogorov-Smirnov statistics for each term. A *p*-Value cut-off of <0.05 was used to evaluate significant GO term enrichment.

### Yeast two-hybrid (Y2H) and DPNR
SIEC-Arabidopsis protein interactions were identified by yeast two-hybrid (Y2H) analysis and mapped as described in Altmann et al., 2020[32], Mukhtar et al., 2011[30] and Wessling et al., 2014[11]. Full details can be found in the supplementary information of these articles. All 106 SIECs were screened against the Arabidopsis ORF library reported in Wessling et al., 2014[11], as well as an additional 500 plant hormone proteins[32]. Network analysis was performed using Cytoscape (v3.9.1) and R studio (v3.1). All network graphs were created using tools in the base Cytoscape program. For intra- and interspecies convergence analysis, degree preserved network rewiring (DPNR) was used. For intraspecies convergence, *N* genes were sampled randomly with replacement from the AI1-MAIN network[31], where *N* was the number of total PPIs between microbial effectors and host proteins observed by Y2H. The total number of host proteins sampled from one simulation

were stored and this process was repeated 10,000 times. Significance was calculated by dividing the number of simulations where the calculated number of targets was fewer than the number of targets observed by Y2H. If the simulated number was never lower than the observed number by the simulations, the $p$-Value was set to <0.001. For interspecies convergence, performed between $Si$ and the five pathogens, $N$ genes were sampled randomly with replacement from the AI1-MAIN network, where $N$ was the total number of unique targets to each set of effectors. The number of common interactors between samplings was stored and this simulation was repeated 10,000 times. Significance was calculated by dividing the number of simulations where the calculated number of common interactions was higher than the number observed by Y2H. If the calculated number was never higher than the observed number, the $p$-Value was set to <0.001.

### Yeast signal sequence trap (YSST)

To confirm secretion of identified SIECs in vivo, we used the yeast signal sequence trap (YSST) system as described by Krijger et al., 2008[29]. For yeast transformations we used the pSMASH vector, and generated fusions of full-length SIECs containing their native signal peptide with the yeast invertase SUC2. Growth of yeast cultures on media with only sucrose as carbon source indicates SUC2 secretion due to a functional effector signal peptide.

### Co-immunoprecipitation

For co-immunoprecipitation assays, SIECs were subcloned into pEarleyGate104[81] and thus fused N-terminally with YFP. Putative Arabidopsis targets of SIECs were N-terminally tagged with a FLAG-tag by subcloning them into pEarleyGate202[81]. Leaves of 4-week-old *Nicotiana benthamiana* plants were co-infiltrated with *A. tumefaciens* (GV3101) containing respective constructs for SIECs and Arabidopsis targets, as well as the p19 silencing repressor[87]. After 3 days, leaf tissues were ground and frozen. Proteins were extracted in extraction buffer [150 mM Tris-HCl pH 7.5, 150 mM NaCl, 5 mM EDTA, 2 mM EGTA, 5% (v:v) glycerol, 0.2% (w:v) polyvinylpyrrolidone, 1% (v:v) IGEPAL® CA630, 10 mM dithiothreitol (DTT), 1% (v:v) Plant Protease Inhibitor Cocktail (Sigma), 0.5 mM phenylmethylsulfonyl fluoride (PMSF)] and centrifuged at 30,000 g, 4 °C for 25 min. Samples were incubated with GFP-Trap® affinity matrix (gta-10, Chromotek, Alpaca, nanobody) for 3 h. Affinity matrix was washed 5 times using wash buffer [150 mM Tris-HCl pH7.5, 150 mM NaCl, 5 mM EDTA, 2 mM EGTA, 5% (v:v) glycerol, 10 mM DTT, 0.5% (v:v) Plant Protease Inhibitor Cocktail (Sigma), 0.5 mM PMSF]. Samples were separated using SDS-PAGE and subsequently analysed by Western blot. SIECs were detected using an α-GFP-HRP antibody (sc-9996, Santa Cruz Biotechnology, mouse, monoclonal, 1:10,000), Arabidopsis proteins were detected using α-FLAG ((F3165, Merck, mouse, monoclonal, 1:2000) and α-mouse-HRP (71045, Merck, Goat, polyclonal, 1:10,000) antibodies.

### Arabidopsis genotyping

For genotyping of Arabidopsis mutants, one leaf per individual plant was flash frozen in liquid $N_2$ and ground to a fine powder with metal beads using a TissueLyser (Qiagen). Leaf material was incubated under constant mixing for 10 min in 500 μl of DNA extraction buffer [200 mM Tris-HCl (pH 7.5), 250 mM NaCl, 25 mM EDTA, 0.5% (v/v) SDS], and then centrifuged for 10 min at 13,000 g. The supernatant was transferred into 500 μl chloroform and mixed for 5 min. After centrifugation for 10 min at 13,000 g, the supernatant was transferred into 500 μl isopropanol and let to precipitate for at least 2 h at −20 °C. After centrifugation for 10 min at 13,000 g, the supernatant was removed, and the pellet washed with 70% ethanol. After centrifugation, the supernatant was removed, the pellet dried completely and dissolved in $H_2O$. 100 ng of genomic DNA served as template in PCR reactions with a standard *Taq* polymerase in a thermo cycler using a standard PCR program. Primers are listed in Supplementary Data 7.

### Gene expression analysis by quantitative real time-PCR

Whole seedlings were flash frozen and ground to a fine powder. Total RNA was extracted using TRIzol® reagent (Invitrogen). 2 μg RNA were digested with DNAse I (Thermo Fisher Scientific) in the presence of RiboLock Rnase Inhibitor (Thermo Fisher Scientific) to remove genomic DNA. cDNA synthesis was performed using the qScript™ cDNA Synthesis Kit (Quantabio) according to the manufacturer's instructions. qRT-PCR was performed using the SYBR® Green JumpStart™ *Taq* ReadyMix™ (Sigma) following a standard protocol. The $2^{-\Delta Ct}$ and $2^{-\Delta\Delta Ct}$ methods[88] were used to determine absolute and relative gene expression, respectively. Primers are listed in Supplementary Data 7.

### Protoplast screening

Protoplast screening was conducted as described in Lehmann et al., 2020[49]. Arabidopsis mesophyll protoplasts were generated from the leaves of 4-5-week-old Col-0 plants. 3-4 leaves from 24 plants were sliced into 1 mm strips and incubated in 3 ×6 ml enzyme solution [20 mM MES (ph 5.7), 400 mM mannitol, 20 mM KCl, 1.5% (w:v) cellulase R10 (Melford Laboratories Ltd., C8001), 0.4% (w:v) macerozyme R10 (Melford Laboratories Ltd., M8002), 10 mM CaCl₂, 0.1% BSA] for 2.5–3 h at 25 °C with gentle shaking. Protoplast suspensions were filtered through a 70 μm nylon cell strainer and spun for 2 min at 100 g at 4 °C. Protoplasts were resuspended in W5 buffer [2 mM MES (pH 5.7), 154 mM NaCl, 125 mM CaCl₂, 5 mM KCl) and spun again under the same conditions. Pellets were resuspended in MMG buffer [4 mM MES (pH 5.7), 400 mM mannitol, 15 mM MgCl₂]. Protoplasts were transformed in 96 well microtiter plates or manually in tubes using 3 μg of DNA per 10,000 protoplasts; 1 μg *35S::SIEC*, 1 μg *pHORMONE::LUC* (ABA marker: *pRD29a::LUC*, AUX marker: *pGH3.3::LUC*, JA marker: *pJAZ10::LUC*, SA marker: *pWRKY70::LUC*, CK marker: *pARR6::LUC*) and 1 μg *pUB-Q10::GUS* as internal control for transformation efficiency. Plates were incubated in a growth chamber overnight (22˚C, 12 h light period). Following overnight incubation, 100 μl of supernatant were removed from each well before cells were treated with mock (0.05% EtOH, 0.05% DMSO or water), 50 μM MeJA, 30 μM SA, 500 nM NAA (AUX), 10 μM ABA or 20 μM t-zea (CK). Plates were mixed by gentle shaking for 1 min and incubated for 3–5 h in a growth chamber. For quantification of LUC activity, 20 μl of luciferase substrate [1 mM beetle luciferin, 3 mM ATP, 15 mM MgSO₄, 30 mM HEPES (pH 7.8)] were added to white 96-well plates (NUNC U96, Greiner) before treated protoplasts were transferred using cut tips. Plates were incubated in the dark for 15–30 min and then imaged using a Photek camera system or Tecan plate reader Infinite® M Plex. Photon integration was performed for 15 min – 1 h with the Photek camera or 1000 ms measured over a period of 30 min with the Tecan plate reader. After quantification of LUC activity, 90 μl of supernatant were removed from each well. 100 μl of lysis solution [25 mM Tris/H₃PO₄ (pH 7.8), 2 mM DACTAA, 2 mM DTT, 10% (v:v) glycerol, 1% (v:v) Triton X-100] were added at room temperature. Plates were shaken at 1,000 rpm for 5 min and spun at 1,000 g for 2 min. 10 μl cell lysate was mixed with 100 μl GUS substrate [10 mM Tris-HCl (pH 8), 1 mM 4-methylumbelliferyl-beta-D-glucuronide (MUG), 2 mM MgCl₂] in a black flat bottom 96 well plate and incubated at 37 °C for 1–1.5 h. GUS activity was measured by fluorescence excitation at 360 nm and detection at 465 nm. For analysis, LUC values were normalised using the GUS activity fluorescence value as a control for transformation efficiency. Wells with a GUS activity <50% of the highest 10% GUS fluorescence value on the same plate were omitted. Two technical replicates for each SIEC-marker combination were used to evaluate mock and basal LUC activity, respectively. Integration images were quantified using Image32 (Photek). To select effector candidates that trigger robust changes in phytohormone signalling, we calculated a ranking factor by multiplying the effector/ empty vector ratios for mock and hormone-treated samples. This preprocessing prioritises effectors which have a strong effect on the marker, particularly those candidates that change both basal and

induced expression of the marker in the same manner (either induction or suppression). Following a log2 conversion, the absolute values were ranked from largest to smallest and the Top 10 effectors in each hormone pathway were used for repetitions of protoplast transfections to confirm the observed changes on the respective markers. For evaluation of *pHORMONE::LUC* activity in T-DNA insertion mutants, GUS-normalised LUC values were normalised to the Col-0 mock value to reduce variation between biological replicates. Data were analysed by two-way ANOVA followed by a Tukey test.

## Hormone tolerance assays

Arabidopsis *35S::SIEC* lines were phenotyped for growth promotion and hormone tolerance by comparing their mock vs treated % root reduction to that of control plants (*35S::GFP*). T-DNA insertion lines of SIEC targets were phenotyped by comparing them to Col-0. Seeds were surface sterilised and plants were grown for seven days on ½ MS media after stratification in the dark at 4 °C for 48 h. SIEC-expressing seedlings were selected with 10 µg ml$^{-1}$ BASTA. Seedlings were then transferred to mock plates (½ MS without hormone) or ½ MS plates supplemented with a respective hormone (10 µM ABA, 40 nM IAA, 200 nM BA, 10 µM SA, or 0.5 µM MeJA). After seven days, photos of plates were taken and root lengths and lateral root numbers were measured using ImageJ. Hormone tolerance was quantified by calculating the relative change in root length/LRN between *35S::SIEC* lines grown on treatment vs mock plates. These values were then compared to the relative change in *35S::GFP* treated in the same way, by dividing the *35S::SIEC* % change by the mean relative % response of *35S::GFP* control lines, to calculate the percentage difference in responsiveness. Tolerance was then calculated by subtracting 100, where negative values indicate reduced tolerance, and positive values indicate increased tolerance. For anthocyanin quantification[89], seeds were surface sterilised, stratified for 48 h, and SIEC-expressing seedlings selected on BASTA containing ½ MS media after 7 days in short day conditions. 14 GFP and SIEC expressing plants per plate were then transferred to media containing 0 nm, 200 nM, 500 nM, 1 µM and 10 µM CK and kept in continuous light. After 7 days, seedlings were harvested in liquid N$_2$. After grinding, 5 volumes of anthocyanin extraction buffer (45% methanol, 5% acetic acid) per mg fresh weight were added. Samples were centrifuged at 12,000 g for 5 min and the supernatant was transferred to a new tube. This step was repeated, and 200 µl of the final sample transferred into transparent V-bottom 96-well plates. Absorbance was measured at 530 and 637 nm using a Tecan Infinite® M Plex plate reader. Relative anthocyanin content was calculated by [Abs530 − (0.25 × Abs657)] × 5. The experiment was repeated 3 times using 2 technical repeats each time.

ABA tolerance was determined by evaluating seed germination following ABA treatment. Seeds were surface sterilised and plated on ½ MS media (mock) and ½ MS media containing 0.1 µM ABA in 6 rows with about 20 seeds per row. After stratification for 48 h at 4 °C in the dark, the plates were transferred into short day conditions for 2 days. Germinated seeds per row were counted, and germination rates determined. The experiment was repeated 3 times.

## Quantification and statistical analysis

All experiments were completed at least three times except where indicated in the figure legend. In these instances, the experiment would be based on at least two biological replicates. Statistical analyses of the RNAseq data and networks, including the integrated 12k_space cross-kingdom interactome map and FI-SPIN are detailed above. For comparative analyses, *Si* interactions identified in the 500 hormone gene search space were omitted to reduce bias. Luciferase activity data from protoplast assays was processed and analysed as described above. Due to large variation of all samples between biological replicates, luciferase activity values were normalised to Col-0 mock. The

normalised values from at least two biological replicates were then used for one- or two-way analysis of variance (ANOVA) with Tukey's multiple comparison test, considering genotype, treatment and interaction, where applicable. For phenotyping assays, significance was determined using a two-tailed, unpaired *t*-test on either the absolute root length/lateral root number, or for hormone assays the percentage difference to the mock treated samples. For YSST assays two-tailed, unpaired *t*-test was used on either absolute OD$_{600\ nm}$ or the percentage difference to samples containing *pSMASH* without signal peptide. We used the function runTest to calculate both Fisher and Kolmogorov–Smirnov statistics for each term

## Reporting summary

Further information on research design is available in the Nature Portfolio Reporting Summary linked to this article.

## Data availability

All functional, transcriptional, genetic and interaction data is available in Supplementary Data 1–8. The transcriptional analysis of *Si* colonised Arabidopsis roots and SIEC identification, including DEGs in host tissues at 3 and 10 dai are available in Supplementary Data 1. The interactions of SIECs and Arabidopsis proteins identified through Y2H, as well as published pathogen effector interaction data and the search spaces used here are presented in Supplementary Data 2. All GO enrichment results for the SIEC interactome in the 8k and 12k spaces, exclusive and shared SIEC targets, and DEGs at 10dai are presented in Supplementary Data 3. Significantly enriched GO terms were calculated by Fisher and Kolmogorov-Smirnov tests with, a p value cut-off of 0.05. The classification of SIEC target proteins as exclusive to *Si* or shared with a pathogen effector in the 8k and 12k are presented in Supplementary Data 4. The initial and confirmation screens of SIEC function in protoplasts against *pHORMONE::LUC* constructs are presented in Supplementary Data 5. All physiological data for TDNA and *35 S::SIEC* line performance in phenotyping assays and outcomes of two-sided, unpaired t-tests are presented in Supplementary Data 6. All primers used in this study are listed in Supplementary Data 7. RNA sequencing data is deposited with GEO under accession code GSE222356. Source data for all figures are provided with this research article. Read counts mapped to TAIR10 are available in Supplementary Data 8. Source data are provided with this paper.

## Code availability

Scripts for performing DPNR in R are available at https://doi.org/10.5281/zenodo.7749043[90].

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

## Acknowledgements

We thank Jens Steinbrenner for critical comments on the manuscript. This work was funded by research grants from Biotechnological and Biological Research Council (BBSRC) / Engineering and Physical Sciences Research Council Grant (EPSRC) of the United Kingdom (BB/M017982/1) and BASF Plant Science Company GmbH.

## Author contributions

Research concept and design: S.L., P.F.-B. and P.S. SIEC-At interactome mapping, data integration and convergence statistics: R.O., M.A., S.A. and P.F.-B. Immunoprecipitation and yeast secretion assays: R.O. and L.R. Si-At RNAseq analyses: R.E. and C.R.-G. Protoplast assays: L.R., E.K. A.D.-F. and S.L. Generation of SIEC lines and phenotyping: R.O., L.R., S.L., E.K., J.R., Y.Z., E.O., C.S. and W.S. Mutant genotyping and phenotyping: L.R. and E.K. Manuscript writing: R.O., L.R., S.L., P.F.-B. and P.S. Critical manuscript reading and editing: R.E., V.N. and W.S.

## Funding

## Competing interests

The authors declare no competing interests.
