## [Peer Review File · Nature Communications]

Symbiont-host interactome mapping reveals effector-targeted modulation of hormone networks and activation of growth promotionReviewers' Comments:

Reviewer #1:

Remarks to the Author:

This manuscript generated a large-scale interaction map between effectors of the beneficial fungus *Serendipita indica* (Si) and Arabidopsis proteins and revealed host effector targets, common to and distinct from pathogen effectors.

The authors screened Arabidopsis proteins interacting with 106 Si effectors in yeast and identified candidate host proteins, followed by confirmation with in planta protein-protein interaction assays. Then, they compared this interaction map with previously studied ones for Arabidopsis and pathogen effectors. Host targets by Si and pathogen effectors are significantly overlapped, indicating the convergent targeting strategy by non-pathogen and pathogens to infect plants. Si effectors targeted phytohormone pathways more frequently than random and pathogen effectors, showing distinct functions of Si effectors compared with pathogen effectors. Then, they showed that some of Si effectors actually interfere with phytohormone pathways. They also showed that host effector targets, previously unrelated to phytohormone pathways, were involved in phytohormone signaling.

How the consequence of an interaction between the host and a microbe is determined is an important, fundamental question. In particular, how a microbe becomes beneficial or pathogenic to the host is currently a hot topic. This research showed that beneficial fungus and pathogens target common host proteins. Furthermore, Si effectors also specifically target host proteins. Their findings will pave the way toward understanding how a microbe becomes beneficial or pathogenic. This study provided massive data and several conceptual advances. While this study did not reveal mechanistic details about how Si effectors modulate the host to promote plant benefit, it addressed important questions by using large-scale experiments and data analyses. The manuscript is written well. Nevertheless, there are several points that the authors should clarify to further improve this study.

Major comments

1. Title: The authors showed that some of Si effectors promote root growth. I wonder if this is truly beneficial to the host. The authors may use something like promotion of plant growth instead of host benefit. Related to this, as this is the major point, FigS6C should be in the main figure. These are suggestions.

2. Fig4F-K and Fig5D-I: It is unclear how the authors calculated the values. In Method, it sounds like they compared differences in responses to hormones but the Y-axis says root length relative to control, which confused me. In addition, 35S::SIEC10 showed a shorter root compared with the control GFP and other 35S::SIEC lines in FigS6C. This 35S::SIEC10 line is the one showing less tolerance (more sensitive?) to ABA, AUX, and JA. I wonder if this shorter root of 35S::SIEC10 affected the results in Fig4. In any case, the authors should describe these as clearly as possible because this is a major point of this study. The authors also used two terms, "sensitivity" and "tolerance" in the text. To increase the readability, the authors should use one consistent term. Personally, I can more easily understand if they use sensitivity throughout the manuscript.

3. FigS4: The authors cannot confirm overexpression of SIEC in this way as there should be no SIEC expression in the GFP line. The detected RT-qPCR signals should come from non-specific or primer-dimer amplification. As this is not so important, the authors may delete this figure and related text. Alternatively, the authors can measure GFP expression in the GFP line. Probably, there is no convenient way to confirm overexpression of foreign genes as there is no endogenous control. But showing GFP expression in the GFP control would provide clues to a certain degree.

Minor comments

1. Fig1B: The authors should explain what "PPI" is.

2. Fig1E (SIEC21): There should be a band for only YFP-SIEC21 in the α -GFP panel. In addition, the quality of the α -GFP immunoblotting for SIEC is too low.

3. Fig2D: The right bottom figure should be labeled with Si vs Xcc vs Rps vs Gor according to the text.

4. Fig3C: The light blue color for the dots is the same as ABA in Fig3A/B. This might confuse readers. This is also applied to Fig5B.
5. L119-121: DB-SIEC or DB-SIECs. It is better to be consistent.
6. L128 and L130 (L139): "biochemically orthogonal co-immunoprecipitation assays in planta" "biophysical quality" These are uneasy to follow. The authors may simplify these statements.
7. L171: They mentioned TCP9, EBS1, CSN5A, AT4G01090. This should be consistent with Fig2E.
8. L172-173: The statement holds true in terms of the number of common interactors. However, the number of host targets for different microbes is different. The authors may want to describe more carefully.
9. L241: It is unclear how they calculated "31%" for significant changes.
10. L279: 35S::SIEC should be in italics.
11. L470: Tween-20
12. L503: The superscript comma after 2020 31 should be a normal comma.
13. L504 Remove the superscript dot after 2014 11.
14. L507: All network graphs were created by hand. What does it mean? The authors should provide more information.
15. L534: pEarleyGate202 needs a reference.

Reviewer #2:

Remarks to the Author:

Osborne et al.

A main conclusion of the manuscript is that the hormone network is a major target of Si effectors in Arabidopsis. This is an interesting study combining bioinformatic tools with wet-lab experiments. The bioinformatic platform has been established and used earlier for pathogenic fungi and is now extended for comparison with the beneficial fungus Si. However, the identification of the target protein/networks and interpretation of some of the data have to be considered with caution, as outlined below. Some of my concerns should be addressed before the manuscript should be considered for publication.

In general: The text suggests that Si targets preferentially hormone proteins/signaling compounds while this is less obvious for pathogens. Pathogens mainly focus on defense hormone activated processes (JA/SA/ET), whereas beneficial microbes have to establish a balance or induce changes between beneficial effects such as growth and development (AUX, ET, BR, CT) and defense (JA, SA, ET). Therefore, one would expect that the overall number of hormone targets is larger for beneficial microbes when compared to those from pathogens. This should be considered in the discussion. Overall, in the discussion, it should be mentioned that the authors only analyze effector-induced effects on host phytohormone signaling, which is often not so clear.

As mentioned on pp. 118-120, screening was performed with a library of 12,000 proteins plus 500 proteins associated with hormone functions. This could result in an enrichment of hormone-related proteins in the screen. Has this been considered in their analysis?

It is also not very well discussed whether the effectors directly affect hormone functions/signaling, or whether the observed changes in the hormone-related processes are downstream responses in the host to respond to cellular changes which are mediated by the effectors (cf. l. 87, "unique Si targets within the host hormone network" (?), whereas the next sentence is ok: "modulate hormone signaling"), (cf. also l. 255/6, which needs explanation).

l. 93-95: very general statement: Investigated was one beneficial microbe. Sentence should be more precise.

l.131ff: Three GO terms are mentioned, but there are more in the two Tables. The broad spectrum of targets should be mentioned or at least discussed. The main focus and examples for common targets of effectors from the beneficial fungus and pathogenic organisms are defense responses in the manuscript, what about the other targets?

l. 171: I find only TCP9 in Fig. 2E.

l.242 ff and next chapter: the change between % and "number of SIECs/reporter gene" is confusing. Numbers are easier to understand.

p. 256: changes in several hormone marker assays: This obviously shows that the primary targets of the effectors could be anywhere in the plant metabolism with consequences for hormone homeostasis, which could also be the case for the effectors with effects on a single hormone pathway

(cf. above). It is not surprising that microbe-induced reprogramming in plants cause changes in hormone balance. Therefore, the authors should point out what is the new outcome of this analysis. Chapter starting with l. 257: The readouts are quite poor. Hormone data with and without stimuli would support the conclusions. Also the statement in l. 395 (experimentally observed hormone changes) is not true.

l. 281ff: The observation that individual SIEC 35S-lines promoted root growth is clearly nice, but I feel that more information for the lines is required. Which hormone network is affected, and are there other beneficial effects besides root growth promotion/inhibition? Those plants with benefits (Suppl. Fig. 6C) should be at least classified according to the altered hormone networks (l. 281-3).

Abstract: The last statement "stressful conditions" in the Abstract is not really supported by the data. l. 375/6: very general sentence with little information.

Reviewer #3:

Remarks to the Author:

Overview: Pathogen and beneficial microbes make use of the same plants pathways to interact with plants. However, what these exact pathways are, what the beneficial-microbe specific pathways are, and how a plants recognizes a pathogen versus a beneficial microbe is unclear. The author aim to address these questions in a systemic large-scale interaction study. This results in the identification of 106 potential effectors of the beneficial microbe *Serendipita indica* (Si) and their plant interaction partners. The authors compare these interactions with pathogen related effectors and their plant interaction partners to identify the shared and unique pathways from Si. The authors identified hormone-related interactions and tested the Si effectors as well as their interaction partners for a function in hormone-related pathways in protoplasts assays and in planta. I applaud the authors for their thorough and significant research. The manuscript is well-written and brings forward the a systemic study on beneficial microbe and plant interactions.

Major

- The authors conclude that the Si-exclusive host targets are distinctly enriched for GO terms associated with hormone responses. However, in Figure 2G, the shared targets are 3.51 times enriched for the hormone response GO term, while the exclusive targets only 2.52 times. There conclusion about hormones. Please revise/soften the conclusions in the results and discussion regarding the distinct enrichment of GO terms for hormones especially since the hormone interaction network is used as starting point, increasing the changes to detect hormone-related relations.

Additionally, it is not clear for the reviewer how the GO terms on which the authors focus have been picked. There are 219 significant GOs for the exclusive targets with the top GO terms being programmed cell death, phosphorylation, signaling. Please use a systemic approach for GO picking.

- Please describe the DPNR in the methods and provide the script used to perform the DPNR

- It would be a beneficial and helpful if the authors could provide a figure such as a heatmap that shows the correlation/connection between the protoplast hormone induction assays and the in planta data, giving a systemic overview of how well these are correlated. In addition, it is not clear for the reviewer why the authors choose to examine the product effect of the basal additive to the treatment and not look at them separately? Can they add a justification for adding both effects?

Minor

- Line 101: How was the early and late time point chosen? Is there a phenotypic response to *S. indica* already visible ? Please provide all 826 SEIC data as well as all the processed RNAseq data instead of only the DEGs for transparency and easy dissemination.

- Line 157: Please provide a short concise description on how you evaluated intraspecies convergence

Point by point responses

We appreciate the constructive feedback from all reviewers and thank them for taking the time to help improve our manuscript. We have now revised the manuscript according to their suggestions, changes of which have been highlighted. Below is our point by point response to each of the reviewer comments.

Reviewer #1 (Remarks to the Author):

This manuscript generated a large-scale interaction map between effectors of the beneficial fungus *Serendipita indica* (Si) and Arabidopsis proteins and revealed host effector targets, common to and distinct from pathogen effectors.

The authors screened Arabidopsis proteins interacting with 106 Si effectors in yeast and identified candidate host proteins, followed by confirmation with in planta protein-protein interaction assays. Then, they compared this interaction map with previously studied ones for Arabidopsis and pathogen effectors. Host targets by Si and pathogen effectors are significantly overlapped, indicating the convergent targeting strategy by non-pathogen and pathogens to infect plants. Si effectors targeted phytohormone pathways more frequently than random and pathogen effectors, showing distinct functions of Si effectors compared with pathogen effectors. Then, they showed that some of Si effectors actually interfere with phytohormone pathways. They also showed that host effector targets, previously unrelated to phytohormone pathways, were involved in phytohormone signaling.

How the consequence of an interaction between the host and a microbe is determined is an important, fundamental question. In particular, how a microbe becomes beneficial or pathogenic to the host is currently a hot topic. This research showed that beneficial fungus and pathogens target common host proteins. Furthermore, Si effectors also specifically target host proteins. Their findings will pave the way toward understanding how a microbe becomes beneficial or pathogenic.

This study provided massive data and several conceptual advances. While this study did not reveal mechanistic details about how Si effectors modulate the host to promote plant benefit, it addressed important questions by using large-scale experiments and data analyses. The manuscript is written well. Nevertheless, there are several points that the authors should clarify to further improve this study.

We thank the reviewer for this thorough summary of our paper, and their positive feedback. We have made changes to the manuscript and the figures according to their recommendations/comments.

Major comments

1. Title: The authors showed that some of Si effectors promote root growth. I wonder if this is truly beneficial to the host. The authors may use something like promotion of plant growth instead of host benefit. Related to this, as this is the major point, FigS6C should be in the main figure. These are suggestions.

We have now moved Fig. S6C from the supplement to Fig. 4 (now Fig. 4L) to reflect the major points we discuss with respect to SIEC mediated root growth promotion. We have also altered the title of our manuscript to reflect the critical difference between host benefit and growth promotion

2. Fig. 4F-K and Fig. 5D-I: It is unclear how the authors calculated the values. In Method, it sounds like they compared differences in responses to hormones but the Y-axis says

root length relative to control, which confused me. In addition, 35S::SIEC10 showed a shorter root compared with the control GFP and other 35S::SIEC lines in FigS6C. This 35S::SIEC10 line is the one showing less tolerance (more sensitive?) to ABA, AUX, and JA. I wonder if this shorter root of 35S::SIEC10 affected the results in Fig4. In any case, the authors should describe these as clearly as possible because this is a major point of this study. The authors also used two terms, “sensitivity” and “tolerance” in the text. To increase the readability, the authors should use one consistent term. Personally, I can more easily understand if they use sensitivity throughout the manuscript.

We apologize for the confusion. We have revised the materials and methods to describe (lines 695-701) more clearly how the data for Fig. 4F-K and Fig. 5D-I were calculated. The y-axis title has been adjusted to better reflect the performed comparisons in hormone phenotype analyses. With respect to the comments toward line 35S::SIEC10, the initial shorter root growth was considered in the tolerance calculation, as these values reflect the 35S::SIEC10 plant growth in hormone containing, relative to plants grown on mock media without hormones. Nonetheless we have included a new section which reflects how this phenotype may have affected our downstream analysis (line 309-316).

Furthermore, we have now changed all instances of the word ‘sensitivity’ to tolerance for consistency.

Fig. S4: The authors cannot confirm overexpression of SIEC in this way as there should be no SIEC expression in the GFP line. The detected RT-qPCR signals should come from non-specific or primer-dimer amplification. As this is not so important, the authors may delete this figure and related text. Alternatively, the authors can measure GFP expression in the GFP line. Probably, there is no convenient way to confirm overexpression of foreign genes as there is no endogenous control. But showing GFP expression in the GFP control would provide clues to a certain degree.

We agree with the reviewer’s recommendation and performed an additional qRT-PCR for GFP expression in our GFP control plants. In addition, the way we presented the data might have hidden that overexpression was determined relative to two housekeeping genes *UBQ10* and *ERF1 α* for data normalization. We changed the presentation of our data in Fig. S4, and show now SIEC expression in our effector or GFP plants relative to the two housekeeping genes to confirm expression of *Si* effector candidates in Arabidopsis.

Minor comments

1. Fig. 1B: The authors should explain what “PPI” is.

This has now been amended in the MS (line 123).

2. Fig. 1E (SIEC21): There should be a band for only YFP-SIEC21 in the α -GFP panel. In addition, the quality of the α -GFP immunoblotting for SIEC is too low.

We have included a longer exposure blot for YFP-SIEC21 which indicates it has been pulled down in the negative control (second) lane.

3. Fig. 2D: The right bottom figure should be labeled with *Si* vs *Xcc* vs *Rps* vs *Gor* according to the text.

The graph for four-way convergence analysis has now been included as part of supplementary Fig. 2C, and the text in the MS (line 176, Fig. 2 legend) revised to reflect Fig. 2D showing the three-way convergence analysis for *Si* vs *Xcc* vs *Rps*.

4. Fig. 3C: The light blue color for the dots is the same as ABA in Fig. 3A/B. This might confuse readers. This is also applied to Fig. 5B.

We have modified Figs. 3C and 5B to reflect this important change. The light blue colour for ABA was used to allow clarity between the different types of interactions, but the legends describing first and second degree SHIPs and PCPs in Figs. 3C and 5B, respectively, are now 'neutral' so as not to cause confusion.

5. L119-121: DB-SIEC or DB-SIECs. It is better to be consistent.

This has now been amended in the MS.

6. L128 and L130 (L139): "biochemically orthogonal co-immunoprecipitation assays in planta" "biophysical quality" These are uneasy to follow. The authors may simplify these statements.

We have simplified these phrases; thank you (lines 128 and 148)

7. L171: They mentioned TCP9, EBS1, CSN5A, AT4G01090. This should be consistent with Fig. 2E.

We have adjusted Fig. 2E to reflect all target proteins mentioned in the text (line 182-183). This was also mentioned by reviewer 2.

8. L172-173: The statement holds true in terms of the number of common interactors. However, the number of host targets for different microbes is different. The authors may want to describe more carefully.

We have removed the word 'proportionally' here to reflect the reviewers important comment and what we conclude from these data. Indeed the overlap observed was not normalised to the total number of host targets between the three microbes. When this is taken into account, the highest % overlap is with *Gor* effectors (31.6% overlap), while it is 28% overlap to *Rps* effectors. We feel that discussing the biological significance of these small differences is less meaningful than the absolute overlap, and hence we do not mention this further in the manuscript.

9. L241: It is unclear how they calculated "31%" for significant changes.

This has now been described more clearly in the text, using both % and the raw number of changes observed (line 273-275). Given the large number of combinations screened, significant changes in the initial screen were calculated by whether the multiplied ratio in mock and treated conditions was either less than 0.5 or larger than 2.

10. L279: 35S::SIEC should be in italics.

11. L470: Tween-20

12. L503: The superscript comma after 2020 31 should be a normal comma.

13. L504 Remove the superscript dot after 2014 11.

14. L507: All network graphs were created by hand. What does it mean? The authors should provide more information. (line 573)

15. L534: pEarleyGate202 needs a reference.

Comment 10-15: We thank the reviewer for their diligent observations with respect to grammatical and typographical errors throughout the manuscript. These have now been corrected throughout and included ref. 81 for pEarleyGate202.

Reviewer #2 (Remarks to the Author):

A main conclusion of the manuscript is that the hormone network is a major target of Si effectors in Arabidopsis. This is an interesting study combining bioinformatic tools with wet-lab experiments. The bioinformatic platform has been established and used earlier for pathogenic fungi and is now extended for comparison with the beneficial fungus Si. However, the identification of the target protein/networks and interpretation of some of the data have to be considered with caution, as outlined below. Some of my concerns should be addressed before the manuscript should be considered for publication.

We thank the reviewer for taking the time to review our manuscript and for all of their critical feedback. We feel we have addressed their concerns with respect to the interpretation of the data, and where necessary provide further clarity on how particular conclusions were reached.

In general: The text suggests that Si targets preferentially hormone proteins/signaling compounds while this is less obvious for pathogens. Pathogens mainly focus on defense hormone activated processes (JA/SA/ET), whereas beneficial microbes have to establish a balance or induce changes between beneficial effects such as growth and development (AUX, ET, BR, CT) and defense (JA, SA, ET). Therefore, one would expect that the overall number of hormone targets is larger for beneficial microbes when compared to those from pathogens. This should be considered in the discussion.

We thank the reviewer for this pertinent point and have now considered this in the discussion (lines 448-455). Although it is reasonable to assume that pathogens would indeed target fewer hormone targets vs a symbiont due to reduced requirement to modulate growth and developmental effects, we do not feel this is a particularly obvious assumption to make. There is an increasing branch of research demonstrating the requirement of pathogen effectors to target hormone pathways other than SA/JA/ET. In addition, a number of reviews by others and us have indicated the high interconnectivity of hormone pathways and that all hormones have functions in development and stress responses (Pieterse et al. 2012, Ann. Rev. Cell Dev. Biol., doi.10.1146/annurev-cellbio-092910-154055; Reitz et al. 2015 J. Exp. Bot., doi.org/10.1093/jxb/erv106; Vanstraelen & Benkova 2012, Ann. Rev. Cell Dev. Biol., doi.10.1146/annurev-cellbio-101011-155741). Our still limited knowledge about the temporal and spatial resolution of hormonal changes during host-microbe interactions and the high molecular complexity demand a continued research effort. Analysing microbial effector targeting for other mutualistic and pathogenic microbes will help to recognize which patterns are shared between the two groups and which strategies prove to be characteristic for either lifestyle. Because of this, we hope to the reviewers give consent that there is not enough supporting evidence to speculate too much on why we observed increased hormone targeting in the Si effector targets.

Overall, in the discussion, it should be mentioned that the authors only analyze effector-induced effects on host phytohormone signaling, which is often not so clear.

We included the term 'effector' whenever it was less obvious in the discussion.

As mentioned on pp. 118-120, screening was performed with a library of 12,000 proteins plus 500 proteins associated with hormone functions. This could result in an enrichment of hormone-related proteins in the screen. Has this been considered in their analysis?

We thank the reviewer for this very important observation about how the study was conducted. In fact, we had realized this complication and had removed these 500 proteins from all downstream comparative analysis as *Gor*, *Xcc*, *Rps*, *Hpa* and *Psy* effectors were not screened

in this larger search space (line 724-727). All comparative analyses were only done on the commonly investigated search space, i.e. host proteins.

It is also not very well discussed whether the effectors directly affect hormone functions/signaling, or whether the observed changes in the hormone-related processes are downstream responses in the host to respond to cellular changes which are mediated by the effectors (cf. l. 87, “unique Si targets within the host hormone network” (?), whereas the next sentence is ok: “modulate hormone signaling”), (cf. also l. 255/6, which needs explanation).

Indeed there is complexity and it would be important to consider this in the future. We do not consider that all interactions/phytohormone changes are necessarily consequences of direct effector interactions with hormone synthesis or core signalling proteins, and could indeed be based on subsequent downstream activity. Obviously, we can interpret our findings only to such an extent as we know hormone annotations for Arabidopsis proteins. We revised the discussion to address the reviewer’s concern (lines 288-295, 448-452).

L. 93-95: very general statement: Investigated was one beneficial microbe. Sentence should be more precise.

We have addressed this in the updated version of the manuscript (line 94).

I.131ff: Three GO terms are mentioned, but there are more in the two Tables. The broad spectrum of targets should be mentioned or at least discussed. The main focus and examples for common targets of effectors from the beneficial fungus and pathogenic organisms are defense responses in the manuscript, what about the other targets?

We have now discussed the broader enriched gene ontologies between our various comparisons, reflecting the similarities and differences more critically, and commenting further where applicable. As it was unclear whether the reviewer’s comment was with respect to the GO enrichment of the DEGs or the comparative interaction network, we have adapted both (lines 138-145, 192-221).

I. 171: I find only TCP9 in Fig. 2E.

This was also pointed out by reviewer 1. We have updated Fig. 2E to display the names of all common targets discussed in the main body of the text (line 182-183).

I.242 ff and next chapter: the change between % and “number of SIECs/reporter gene” is confusing. Numbers are easier to understand.

This was also pointed out by reviewer 1. We have updated the text which should now be clearer, showing numbers as well as percentage in the main body of the text (line 273-275).

p. 256: changes in several hormone marker assays: This obviously shows that the primary targets of the effectors could be anywhere in the plant metabolism with consequences for hormone homeostasis, which could also be the case for the effectors with effects on a single hormone pathway (cf. above). It is not surprising that microbe-induced reprogramming in plants cause changes in hormone balance. Therefore, the authors should point out what is the new outcome of this analysis.

We agree that it is not always clear where exactly effectors interact with the hormone signalling pathways as only a limited number of Arabidopsis protein targets identified by our yeast-two-hybrid analysis have a known hormone function. This makes our protoplast and whole plant assays so valuable. In this respect it is important to mention, we previously published (Lehmann et al. 2022 Plos One, doi.org/10.1371/journal.pone.0234154) that our hormone markers are highly specific for the individual hormones. As *S. indica* in general and single effector proteins were shown in literature to modulate hormone signalling, the novelty of our

approach is that we identified hormone effects of specific SIECs in a broad range, high throughput analysis (protoplast assay). It needs to be emphasised that *Si* effectors turned out to be highly suitable for assigning previously unknown hormone functions to Arabidopsis proteins. As reviewer 2 points out it is not necessarily surprising to see hormone modulation by effector proteins, our results, however, indicate a very specific hormone function of individual effectors. We show that on the one hand 71 % of our tested SIECs only changed one or two hormone markers while on the other hand only 6 out of 106 have broad range (inducing or repressing) effect on the hormone markers. These 6 might have extensive effects on different mechanisms of plant metabolism and hormone homeostasis. Therefore, the specifically assigned SIEC-hormone relations offer new mechanistic insights. In addition, our comparative interactome network map revealed Arabidopsis (hormone) targets shared between pathogen and *Si* effectors and those that are unique for the symbiont effectors. All these aspects were previously unknown and are now described/ discussed more clearly in the manuscript (lines 138-145, 192-221).

Chapter starting with l. 257: The readouts are quite poor. Hormone data with and without stimuli would support the conclusions. Also the statement in l. 395 (experimentally observed hormone changes) is not true.

We apologize for this confusion and imprecise description of our analyses. All protoplast and phenotyping assays were done comparatively with and without hormone treatments (in the absence/presence of the effector). The data in Fig. 4A-K and Supplementary Tables 5 and 6 are therefore SIEC effects relative to indicated hormone treatments. The protoplast screen provides quantitative data about effector effects on hormones with or without individual hormone treatments *in planta*. Our approach was to demonstrate that such *in planta* changes translate into altered hormone-based phenotypes in whole plants taking root growth as a readout. By combining quantitative effects of effectors on individual hormones (based on highly specific hormone markers) in our protoplast assay (Fig. 4A-D) with effector effects on hormone tolerance in whole plant assays (Fig. 4E-K), is in our opinion the most appropriate way to indicate that effector functions are specifically associated with hormones.

Line 395 was changed to: observed hormone signalling changes (line 458).

l. 281ff: The observation that individual SIEC 35S-lines promoted root growth is clearly nice, but I feel that more information for the lines is required. Which hormone network is affected, and are there other beneficial effects besides root growth promotion/inhibition? Those plants with benefits (Suppl. Fig. 6C) should be at least classified according to the altered hormone networks (l. 281-3).

We have now included data on the hormone function of SIECs that promote root growth (Fig. S6), which is discussed in lines 325-340. In addition to the root growth promoting phenotype data, we have included phenotypic data on hypocotyl length as a growth-related shoot trait. Please see Fig. 4M.

Abstract: The last statement “stressful conditions” in the Abstract is not really supported by the data.

We have removed this statement from the abstract.

l. 375/6: very general sentence with little information.

We thank the reviewer for their comment. We have edited the passage in the resubmitted version of the manuscript. (line 427-431).

Reviewer #3 (Remarks to the Author):

Overview: Pathogen and beneficial microbes make use of the same plants pathways to interact with plants. However, what these exact pathways are, what the beneficial-microbe specific pathways are, and how a plants recognizes a pathogen versus a beneficial microbe is unclear. The author aim to address these questions in a systemic large-scale interaction study. This results in the identification of 106 potential effectors of the beneficial microbe *Serendipita indica* (Si) and their plant interaction partners. The authors compare these interactions with pathogen related effectors and their plant interaction partners to identify the shared and unique pathways from Si. The authors identified hormone-related interactions and tested the Si effectors as well as their interaction partners for a function in hormone-related pathways in protoplasts assays and in planta. I applaud the authors for their thorough and significant research. The manuscript is well-written and brings forward the a systemic study on beneficial microbe and plant interactions.

We thank the reviewer for their comments, and the wider feedback about the way the analyses were conducted. We feel having addressed their comments that our conclusions are more supported by the data, and that overall the paper is improved thanks to their reporting.

Major

- The authors conclude that the Si-exclusive host targets are distinctly enriched for GO terms associated with hormone responses. However, in Figure 2G, the shared targets are 3.51 times enriched for the hormone response GO term, while the exclusive targets only 2.52 times. There conclusion about hormones. Please revise/soften the conclusions in the results and discussion regarding the distinct enrichment of GO terms for hormones especially since the hormone interaction network is used as starting point, increasing the changes to detect hormone-related relations.

We thank the reviewer for their comments about the GO enrichment in Fig. 2G, their comments about 'response to hormone' being enriched in both shared and exclusive targets, and the selection of GO terms presented/discussed. We conducted a complete reanalyses of enrichment analysis with an updated GO term version. The text in the manuscript, as well as Fig. 2G and the new comparative GO analysis are revised in the updated version of the paper to address the reviewers concern. Please see lines 195-215.

Additionally, it is not clear for the reviewer how the GO terms on which the authors focus have been picked. There are 219 significant GOs for the exclusive targets with the top GO terms being programmed cell death, phosphorylation, signaling. Please use a systemic approach for GO picking.

We now describe the systematic strategy we implemented to select these particular GO terms which we summarise here: GOs were filtered based on the % of the total reference set of genes which was annotated for each term, and then the overall % of the query set which were significant. This way we eliminate a many number of terms which are largely speaking redundant (gene expression, response to stimulus) but also capture a broader sense of the function of each gene set by removing saturation by redundant terms.' Please see section beginning line 198-201 and 556-566.

- Please describe the DPNR in the methods and provide the script used to perform the DPNR

DPNR was initially included in the methods section under the heading 'Y2H and network analysis'. We have now renamed this section 'Y2H and DPNR'. Furthermore, the R script used to perform DPNR has been uploaded at Zenodo and can be accessed at <https://doi.org/10.5281/zenodo.7749043>. Please see lines 567 and 747.

- It would be a beneficial and helpful if the authors could provide a figure such as a heatmap that shows the correlation/connection between the protoplast hormone induction assays and the in planta data, giving a systemic overview of how well these are correlated.

We thank the reviewer for requesting this clarity on the correlation between the hormone marker assays and the phenotyping data. We have now included five heatmaps for the lines which were tested for respective hormone tolerance, with their performance in the protoplast assays. Please see Fig. S6. This new figure is discussed in lines 313-317.

In addition, it is not clear for the reviewer why the authors choose to examine the product effect of the basal additive to the treatment and not look at them separately?

We apologise for this having been unclear in the initial submission. The basal and treated values were multiplied to identify the effectors with the most robust effect on respective pathways. Given that all 106 SIECs were tested against all 5 markers in mock and basal conditions (1060 potential paths of further investigation), we needed to employ some processing step to 'filter' effectors with the strongest effect for each marker. This added robustness to the data and gave us more confidence in our downstream analysis, and helped to integrate effectors with low-medium effects in mock or treated conditions, or remove those with opposite effects on marker responsiveness in mock vs treated conditions. We have updated the main text and the methods section of the paper to reflect why we performed this step (lines 263-273 and 674-679).

Minor

- Line 101: How was the early and late time point chosen? Is there a phenotypic response to *S. indica* already visible ?

These time points were selected as they are considered part of the canonical biphasic infection timepoints for *S. indica* as published by us (Deshmukh et al. 2006, doi.org/10.1073/pnas.0605697103; Jacobs et al. 2011, doi.org/10.1104/pp.111.176446). At day 3 (early), *Si* has considerably proliferated throughout root tissues, and displays a biotrophic colonisation strategy. At day 10, *Si* triggers a localised 'cell death' like response in some colonised cells, effectively replacing those initially biotrophically colonised epidermis or cortex cells. The current hypothesis regarding this biphasic colonisation strategy postulates that *Si* deploys effectors to first repress pattern triggered immunity, and once established deploys an additional suite of secreted proteins required for its proliferation in roots. Phenotypes associated with *Si* colonisation can be seen as early as 3 days after initial infection. Therefore, to capture both sets of secreted proteins in our RNA sequencing, we selected day 3 and day 10 for our study. We have now referenced/explained this more clearly in the manuscript (line 101-103).

Please provide all 826 SEIC data as well as all the processed RNAseq data instead of only the DEGs for transparency and easy dissemination.

The read counts for all Arabidopsis genes in each condition were included in the submission to the GEO. However, in the interest of easier dissemination, we have reformatted this data which is now part of Supplementary Table 1.

- Line 157: Please provide a short concise description on how you evaluated intraspecies convergence

This has now been included in the resubmitted version of the MS described fully in the methods (line 167-168 and 575-581).

Reviewers' Comments:

Reviewer #1:

Remarks to the Author:

The authors have mostly addressed my previous comments, but there are two remaining issues. The first one is related to my previous comment 2. It is still unclear how the authors calculated the values. Why do they need to subtract 100? Can they show simpler values? The second point is related to FigS4. The GFP expression in Col-0 plants and SIEC expression in GFP transgenic plants were technical background noises as they do not have any genuine expression. Therefore, the authors should simply delete those values.

Reviewer #3:

Remarks to the Author:

The reviewer is satisfied with the revisions, including the additional GO analysis, added clarifications in the methods, and increased availability of the source code/data, that the authors have implemented to address the concerns raised. In general, this study provides interesting findings on the systemic response upon beneficial microbe and plant interactions, and it can be considered for publication in this suitable journal.

Minor:

- Please confirm that the hormone markers for the 5 hormones are described in the methods.

Point by point responses

We thank the reviewers for the constructive feedback and for taking the time to improve our manuscript further. We have now made amendments according to the suggestions, which have been highlighted. Below is our point by point response to the comments.

Reviewer #1 (Remarks to the Author):

The authors have mostly addressed my previous comments, but there are two remaining issues. The first one is related to my previous comment 2. It is still unclear how the authors calculated the values. Why do they need to subtract 100? Can they show simpler values?

We thank the reviewer for taking the time to further improve our manuscript. The matter of calculating hormone effects was chosen this way to visualize actual differences between 35S::SIEC and GFP plants in percentual difference of hormone response, while taking 35S::SIEC root growth effects in mock conditions into account. Therefore, our graphs picture the observed differences in the most accurate way.

The calculation was as follows:

35S::SIEC root length change (control mean/treated) in % *100 / mean of GFP root length change (control mean/treated) in % -100

For example if the root length of GFP plants was 75 % in treatment compared to control and the 35S::SIEC line had only 45 % root length on treatment compared to control, the percentual difference between GFP and 35S::SIEC is 40 % ($45 \times 100 / 75 = 60$ % SIEC response is 60 % of GFP response so a reduction of 40 %)

100 was subtracted to simplify differences as positive values indicate increased tolerance, while negative values show reduction of hormone tolerance.

The second point is related to FigS4. The GFP expression in Col-0 plants and SIEC expression in GFP transgenic plants were technical background noises as they do not have any genuine expression. Therefore, the authors should simply delete those values.

Regarding the second point FigS4, we agree to the reviewers suggestion and deleted the GFP or Col-0 background part of the graphs as no expression or only background noise was observed.

Reviewer #3 (Remarks to the Author):

The reviewer is satisfied with the revisions, including the additional GO analysis, added clarifications in the methods, and increased availability of the source code/data, that the authors have implemented to address the concerns raised. In general, this study provides interesting findings on the systemic response upon beneficial microbe and plant interactions, and it can be considered for publication in this suitable journal.

Minor:

- Please confirm that the hormone markers for the 5 hormones are described in the methods.

We thank the reviewer for this summary and help in improving our manuscript. We now made sure all 5 hormone protoplast markers, including reporter gene promoters that were used are mentioned in the methods part of the protoplast assay. Changes are highlighted in the manuscript.